# PDUDT: Provable Decentralized Unlearning under Dynamic Topologies

Jing Qiao [1 2]   Yu Liu [1]   Zengzhe Chen [1]   Mingyi Li [1]   Yuan Yuan [3 4]   Xiao Zhang [1]   Dongxiao Yu [1]

## Abstract

This paper investigates decentralized unlearning, aiming to eliminate the impact of a specific client on the whole decentralized system. However, decentralized communication characterizations pose new challenges for effective unlearning: the indirect connections make it difficult to trace the specific client's impact, while the dynamic topology limits the scalability of retraining-based unlearning methods. In this paper, we propose the first **P**rovable **D**ecentralized **U**nlearning algorithm under **D**ynamic **T**opologies, called PDUDT. It allows clients to eliminate the influence of a specific client without additional communication or retraining. We provide rigorous theoretical guarantees for PDUDT, showing it is statistically indistinguishable from perturbed retraining. Additionally, it achieves an efficient convergence rate of $\mathcal{O}(\frac{1}{T})$ in subsequent learning, where $T$ is the total communication rounds. This rate matches state-of-the-art results. Experimental results show that compared with the Retrain method, PDUDT saves more than 99% of unlearning time while achieving comparable unlearning performance.

## 1. Introduction

With the surge in data volume and increasing geographic dispersion of data sources, some collaborative learning paradigms, such as Federated Learning (McMahan et al., 2017) and Decentralized Learning (Lian et al., 2017), have attracted widespread attentions. In the above collaborative scenarios, privacy regulations like GDPR (Voigt & Von dem Bussche, 2017) grant clients the right to withdraw the use

---

[1]School of Computer Science and Technology, Shandong University, Qingdao, China [2]Zhongtai Securities Institute for Financial Studies, Shandong University, Jinan, China [3]School of Software, Shandong University, Jinan, China [4]Joint SDU-NTU Centre for Artificial Intelligence Research (C-FAIR), Shandong University, Jinan, China. Correspondence to: Yuan Yuan <yyuan@sdu.edu.cn>, Xiao Zhang <xiaozhang@sdu.edu.cn>.

*Proceedings of the 42$^{nd}$ International Conference on Machine Learning*, Vancouver, Canada. PMLR 267, 2025. Copyright 2025 by the author(s).

of their personal data in any form. For instance, users might wish to delete location information from navigation apps or ask smart voice assistants to "forget" sensitive conversation content. Simply deleting data does not ensure the right to be forgotten, as its influence remains embedded in the collaboratively trained models (Tao et al., 2024). Therefore, several studies have empirically explored ways to ensure the right to be forgotten for collaborative learning, using techniques such as knowledge distillation (Wu et al., 2022a), class-discriminative pruning (Wang et al., 2022), projected gradient ascent (Wu et al., 2022b), or second-order AdaHessian optimizer (Liu et al., 2022).

However, the existing works usually rely on a trustworthy central server for coordination, which is not always guaranteed in real-world scenarios (Qiao et al., 2024). To further remove the dependence on the central server, the researchers began to explore how to achieve efficient unlearning in a full decentralized framework. For example, HDUS (Ye et al., 2024) uses distilled seed models to create erasable ensembles for all clients. Similarly, BlockFUL (Liu et al., 2024a) is a novel framework with a dual-chain structure, comprising a live chain and an archive chain, to enable unlearning in Blockchained FL. Although these studies provide practical solutions for decentralized unlearning, the theoretical performance analysis still lacks in-depth exploration.

Therefore, *we aim to design an efficient decentralized unlearning framework, while also theoretically guaranteeing the effectiveness and soundness of the unlearning process.* To achieve this, we need to address the following challenges: **(1) Indirect connections complicate the impact chain.** In a decentralized system, some clients may not be directly connected to the client initiating the unlearning request, yet they can still be influenced through the information flow. The complexity and unpredictability of model propagation paths make it challenging to accurately trace and mitigate the influence of a specific client. **(2) Dynamic topologies make retraining-based methods infeasible.** The constantly changing topologies among clients pose significant barriers to retraining-based unlearning methods. Clients that have exited the decentralized system are often unreachable, making it infeasible to revert to an earlier training state (Tao et al., 2024) or retrain the model from scratch. This lack of access to previously participating clients undermines the consistency and feasibility of retraining approaches, espe-

*Table 1.* Comparison with some related works

| Algorithm | SL | w/o retrain | Dynamic topology | Theoretical guarantee | | | | |
|---|---|---|---|---|---|---|---|---|
| | | | | Unlearning guarantee | Convergence rate | Unlearning time | | Space overhead |
| | | | | | | Comp. | Comm. | |
| FedEraser (Liu et al., 2021) | × | ✓ | × | - | - | - | - | - |
| FedRemover (Yuan et al., 2024) | × | × | ✓ | - | - | - | - | - |
| FedUnl (Wu et al., 2022a) | × | ✓ | × | - | - | - | - | - |
| HDUS (Ye et al., 2024) | ✓ | ✓ | ✓ | - | - | - | - | - |
| FATS-Unl (Tao et al., 2024) | × | × | ✓ | Exact unlearning | $^1\mathcal{O}\big(\frac{1}{\sqrt{Kb(t_1+T)}}\big)$ | $^2 > \min\{1, \frac{KR}{n}\}\cdot$RT | - | $^3\mathcal{O}(R\cdot\max\{b,d\})$ $^3\mathcal{O}(R\cdot\max\{K,d\})$ |
| FedRecovery (Zhang et al., 2023) | × | ✓ | × | $(\epsilon,\beta)$-machine unlearning | - | $\mathcal{O}(t_1)$ | 0 | $\mathcal{O}(t_1 nd)$ |
| **PDUDT (This paper)** | ✓ | ✓ | ✓ | $(\epsilon,\beta)$-machine unlearning | $\mathcal{O}(\frac{1}{T})$ | $\mathcal{O}(t_1)$ | 0 | $\mathcal{O}(t_1 N_{\max}d)$ |

Note that "-" means no result or not applicable, "SL" means "Severless", "RT" means "Retraining time", "$t_1$" is the round when a client in the learning system issues an unlearning request, and $N_{\max} = \max_{i,t} |N_i^t| \le n$ denotes the maximum number of neighbors over rounds in a decentralized learning system.
For the last three columns, we consider the unlearning time and space overhead on a single node, whether it is a server or a client, that needs to perform the unlearning operations.
[1] The original convergence rate in Tao et al.'s paper is shown as $\mathcal{O}(1/\sqrt{\rho_S MN})$. For easy comparison, in our Table 1, it is further derived from line 2 of Algorithm 1 in their paper. Here, $K$ is the number of clients participating in each round of training, and $b$ is the sample batch size used to calculate the gradient (in our theoretical result, we set $b=1$). Their convergence result is for the total training rounds. Since we have proved the statistical indistinguishability of the unlearning algorithm, we only focus on the convergence behavior after the unlearning operations.
[2] Tao et al.'s result relies on the time it takes to retrain the parameterized neural network, which is often much longer than ours. In their result, "$R$" denotes the total communication rounds, and it holds $R = t_1 + T$ in our paper.
[3] The space overhead is $\mathcal{O}(R\cdot\max\{b,d\})$ for each client and $\mathcal{O}(R\cdot\max\{K,d\})$ for the server.

cially when client participation is voluntary and transient. **(3) Global unlearning performance is difficult to quantify theoretically.** In a decentralized setting, each client must independently "forget" the effects of a specific client locally. However, there is currently no unified metric to evaluate how these local unlearning operations collectively achieve the desired global unlearning effect. Without centralized oversight, ensuring that local actions align to produce the intended global impact remains a significant open challenge.

Along this line, we propose a provable unlearning algorithm, called PDUDT, for the decentralized framework under dynamic topologies. To make the whole system "forget" the unlearned client, each client locally uses its own historical gradient submissions, along with those of its neighbors, to perform unlearning operations. Specifically, we compute a sequence of gradient residual approximations using the expected retraining update rule. At the unlearning moment, we subtract the weighted sum of the corresponding approximations from each client's model, where the weights measure the clients' contributions to the system. This process only involves using saved historical information, with no additional communication and training process, making it readily adaptable to dynamic topology settings. To provide a theoretically rigorous unlearning guarantee, we first derive an upper bound on the difference between the retrained model and the output model of the proposed algorithm from a global perspective. Then, we use the Gaussian mechanism to mask this gap in the parameter space, ensuring statistical indistinguishability. Finally, we analyze the impact of the unlearning models on the subsequent convergence behavior of decentralized learning.

Our main contributions can be summarized as follows:

- To the best of our knowledge, we propose PDUDT, the first provable decentralized unlearning algorithm under

dynamic topologies. It modifies local models using only historical information, eliminating the influence of a specific client. Notably, PDUDT requires no extra communication or neural network retraining, ensuring high efficiency in dynamic settings.

- We provide rigorous theoretical guarantee for the PDUDT algorithm. First, we prove that it is statistically indistinguishable from the perturbed retraining method. Then, we derive that the subsequent learning process converges to the scaled difference among the unlearning models from all remaining clients at a rate of $\mathcal{O}(\frac{1}{T})$, which matches the state-of-the-art results.

- We conduct extensive experiments to verify the performance superiority of the proposed PDUDT algorithm. Specifically, compared with the Retrain method, PDUDT saves over 99% unlearning time, while maintaining outstanding unlearning performance.

## 2. Related Work

Depending on the precision of forgetting, existing unlearning techniques in the collaborative frameworks can be categorized into two types: *Exact Unlearning* and *Approximate Unlearning*.

### 2.1. Exact Unlearning

Exact unlearning requires that the clients in a collaborative system completely remove the influence of a specific client. To achieve this, the model typically needs to be retrained or partially retrained, which may incur high computational costs and time expenses (Liu et al., 2024b; Yuan et al., 2024; Tao et al., 2024). For example, one exact unlearning technique is the leave-one-out retraining approach, where the model is retrained on the complete dataset, omitting the

user's data that needs to be unlearned (Liu et al., 2024b). FedRemover (Yuan et al., 2024) designs a real-time malicious client detection scheme to quickly perform unlearning operations and implement a global model of unlearning in a minimum number of rounds. FATS-Unl (Tao et al., 2024) aims to backtrack to the point before the client who initiates the unlearning request first participates in collaborative training, performing retraining to achieve exact unlearning. While these methods can fully eliminate the influence of specific clients, their implementation faces challenges. First, heavy reliance on retraining results in high computational overhead and time costs (Liu et al., 2024b). Second, due to the dynamic nature of client participation in collaborative training, recalling previous clients for retraining is often impractical (Zhang et al., 2023).

## 2.2. Approximate Unlearning

Approximate unlearning provides a more efficient solution, with methods that are typically faster and more computationally economical than exact unlearning (Liu et al., 2024b). Although it may not completely eliminate the influence of specific clients, approximate unlearning is sufficiently effective for most practical scenarios and meets the necessary requirements. Some works have been proposed to improve the understanding of federated approximate unlearning. FedEraser (Liu et al., 2021) relies on the central server to store historical submissions of each client, which are then calibrated to accelerate the unlearning process. FedRecovery (Zhang et al., 2023) provides a federated unlearning scheme to eliminate the influence of a specific client by removing the weighted sum of gradient residuals from the global model.

Considering that the reliability of central servers in practical applications is often not guaranteed, researchers have begun to focus on implementing unlearning in the decentralized frameworks (Wu et al., 2022a; Ye et al., 2024; Lin et al., 2024). For instance, HDUS (Ye et al., 2024) introduces a decentralized unlearning mechanism that leverages distilled seed models to construct erasable ensembles BlockFUL (Liu et al., 2024a) supports unlearning in Blockchained Federated Learning, with an innovative framework featuring a dual-chain structure. Similarly, Lin et al. (2024) addresses unlearning in privacy-preserving AIGC systems via coded computing, which requires additional storage and reconstruction overhead. While these studies offer practical solutions for decentralized unlearning, a thorough theoretical performance analysis remains underexplored.

In Table 1, we provide a comprehensive comparison of our approach against some related works across multiple dimensions, including implementation architecture, communication patterns, theoretical unlearning guarantees, and efficiency metrics.

## 3. Decentralized Unlearning

### 3.1. Problem Setup

In this paper, we consider a decentralized learning scenario with $n$ clients, denoted as $\mathcal{V} = \{1, \cdots, n\}$. The communication mode in round $t$ is modeled by a doubly stochastic matrix $\mathcal{W}^t = (W_{ij}^t)_{n \times n}$, where $W_{ij}^t > 0$ if client-$i$ and client-$j$ can directly communicate. Specially, we consider a general dynamic scenario where the connections between clients can vary arbitrarily after each round, i.e., the neighboring set $N_i^t = \{j | W_{ij}^t > 0, j \in \mathcal{V}, j \neq i\}$ of client-$i$ and weight matrix $\mathcal{W}^t$ of clients vary with the rounds. All clients participating in the training collaboratively find a solution to the following general learning problem:

$$\min_{\theta \in \mathbb{R}^d} f(\theta) := \frac{1}{n} \sum_{i=1}^n \underbrace{\mathbb{E}_{\xi_i \sim \mathcal{D}_i}[F_i(\theta, \xi_i)]}_{:=f_i(\theta)} \tag{1}$$

In round $t$, each client-$i$ receives gradients from all its neighbors, subsequently updating its model through local training that combines these gradients with its own local information according to the communication matrix:

$$\theta_i^t = \theta_i^{t-1} - \eta \sum_{j=1}^n W_{ij}^{t-1} \nabla F_j(\theta_j^{t-1}, \xi_j^{t-1}) \tag{2}$$

After several rounds of training, a client may submit an unlearning request to withdraw its data consent. At this point, due to privacy regulations like GDPR (Voigt & Von dem Bussche, 2017), the decentralized learning system must remove this client's contribution to the global model. Without loss of generality, assume client-$n$ makes the unlearning request at round $t_1$. Ideally, retraining by the $n - 1$ clients, excluding client-$n$, ensures that the contribution of client-$n$ is fully removed, thus guaranteeing privacy. The retraining process can be expressed as follows:

$$\tilde{\theta}_i^t = \tilde{\theta}_i^{t-1} - \eta \sum_{j=1}^{n-1} \tilde{W}_{ij}^{t-1} \nabla F_j(\tilde{\theta}_j^{t-1}, \tilde{\xi}_j^{t-1}) \tag{3}$$

where we can set the initial model as $\tilde{\theta}_i^0 = \theta_i^0$. Theoretically, the communication matrix $\tilde{\mathcal{W}}^t = (\tilde{W}_{ij}^t)_{(n-1) \times (n-1)}$ can still be a doubly stochastic one. For example, the Metropolis-Hastings method (Awan et al., 2006) can be utilized to generate it. And we denote the neighbor set of client-$i$ ($i \in \mathcal{V} \backslash \{n\}$) in round $t$ as $\tilde{N}_i^t = \{j | \tilde{W}_{ij}^t > 0, j \in \mathcal{V} \backslash \{n\}, j \neq i\}$.

However, retraining from scratch incurs prohibitively high computation and communication costs. Therefore, our goal is to adjust the local model $\theta_i^{t_1}$ for each client that continues training, so that from a global perspective, the adjusted models perform similarly to those retrained by $n - 1$ clients.

---

**Algorithm 1** The Perturbed Retraining Algorithm

---

1: **Input:** The number of clients participating in retraining $n-1$, the initial local parameters $\tilde{\theta}_i^0 = \theta_i^0 \in \mathbb{R}^d$ ($i \in \mathcal{V}\backslash\{n\}$), the round $t_1$ when client-$n$ submits an unlearning request, the step size $\eta$, the privacy budget $\epsilon$, and the confidence parameter $\beta$.

2: **for** client $i = 1, \cdots, n-1$ (In Parallel) **do**

3:      **for** round $t = 1, \cdots, t_1$ **do**

4:          Compute its local gradient $\nabla F_i(\tilde{\theta}_i^{t-1}, \tilde{\xi}_i^{t-1})$;

5:          Receive all the gradients $\nabla F_j(\tilde{\theta}_j^{t-1}, \tilde{\xi}_j^{t-1})$ ($j \in \tilde{N}_i^{t-1}$) from its neighbors;

6:          Update each local model parameter following $\tilde{\theta}_i^t = \tilde{\theta}_i^{t-1} - \eta \sum_{j=1}^{n-1} \tilde{W}_{ij}^{t-1} \nabla F_j(\tilde{\theta}_j^{t-1}, \tilde{\xi}_j^{t-1})$;

7:      **end for**

8:      Set $\sigma = \frac{1}{\sqrt{2}} \cdot \frac{d_1}{\sqrt{\log(1/\beta)+\epsilon} - \sqrt{\log(1/\beta)}}$, where $d_1$ is the upper bound discussed in Theorem 4.9;

9:      Add perturbation to each local model parameter $\tilde{\theta}_i^{t_1} = \tilde{\theta}_i^{t_1} + z_i$, where $z_i \sim \mathcal{N}(0, (n-1)\sigma^2 \mathbb{I}_d)$ is a noise from the Gaussian distribution.

10: **end for**

---

## 3.2. Algorithm Design

When client-$n$ issues an unlearning request in round $t_1$, for any client-$i$ ($i \in \mathcal{V}\backslash\{n\}$) in the decentralized learning system, its local model can be expressed as

$$\theta_i^{t_1} = \theta_i^0 - \eta \sum_{t=0}^{t_1-1} \sum_{j=1}^{n} W_{ij}^t \nabla F_j(\theta_j^t, \xi_j^t) \qquad (4)$$

Correspondingly, if retrained from scratch to round $t_1$, for any client-$i$ ($i \in \mathcal{V}\backslash\{n\}$), it holds that

$$\tilde{\theta}_i^{t_1} = \tilde{\theta}_i^0 - \eta \sum_{t=0}^{t_1-1} \sum_{j=1}^{n-1} \tilde{W}_{ij}^t \nabla F_j(\tilde{\theta}_j^t, \tilde{\xi}_j^t) \qquad (5)$$

To investigate the effect of client-$n$, we can subtract Equation (4) from Equation (5), and have

$$\tilde{\theta}_i^{t_1} - \theta_i^{t_1}$$
$$= -\eta \sum_{t=0}^{t_1-1} \left( \sum_{j=1}^{n-1} \tilde{W}_{ij}^t \nabla F_j(\tilde{\theta}_j^t, \tilde{\xi}_j^t) - \sum_{j=1}^{n} W_{ij}^t \nabla F_j(\theta_j^t, \xi_j^t) \right)$$
$$\qquad (6)$$

To make an adjustment to the local model $\theta_i^{t_1}$ of each client-$i$ ($i \in \mathcal{V}\backslash\{n\}$), we introduce $r_i^t = \eta \sum_{j=1}^{n-1} \tilde{W}_{ij}^t \nabla F_j(\tilde{\theta}_j^t, \tilde{\xi}_j^t) - \eta \sum_{j=1}^{n} W_{ij}^t \nabla F_j(\theta_j^t, \xi_j^t)$ to represent the gradient residual. Calculating $r_i^t$ involves the trajectory dynamics of all neighbors of client-$i$ obtained by retraining. Therefore, it is not feasible to directly adjust

---

**Algorithm 2** Decentralized Unlearning Algorithm PDUDT

---

1: **Input:** The number of clients $n$, the initial local parameters $\theta_i^0 = \theta^0 \in \mathbb{R}^d$ ($i \in \mathcal{V}$), the round $t_1$ when client-$n$ submits an unlearning request, the step size $\eta$, the privacy budget $\epsilon$, and the confidence parameter $\beta$.

2: **for** client $i = 1, \cdots, n$ (In Parallel) **do**

3:      **for** round $t = 1, \cdots, t_1$ **do**

4:          Compute and storage its local gradient $\nabla F_i(\theta_i^{t-1}, \xi_i^{t-1})$;

5:          Receive and storage all its neighbors' gradients $\nabla F_j(\theta_j^{t-1}, \xi_j^{t-1})$ ($j \in N_i^{t-1}$);

6:          Update each local model parameter following $\theta_i^t = \theta_i^{t-1} - \eta \sum_{j=1}^{n} W_{ij}^{t-1} \nabla F_j(\theta_j^{t-1}, \xi_j^{t-1})$;

7:      **end for**

8: **end for**

9: Receive the unlearning request from client-$n$.

10: **for** client $i = 1, \cdots, n-1$ (In Parallel) **do**

11:      Compute the approximations $\{\delta_i^t\}_{t=0}^{t_1-1}$ of the gradient residuals by Equation (7);

12:      Compute the weight $p_i^t$ for each approximation $\delta_i^t$ ($t = 0, \cdots, t_1-1$) according to Equation (8);

13:      Subtract a weighted sum of $\delta_i^t$ from $\theta_i^{t_1}$ to obtain $\bar{\bar{\theta}}_i^{t_1}$ based on Equation (9);

14:      Set $\sigma = \frac{1}{\sqrt{2}} \cdot \frac{d_1}{\sqrt{\log(1/\beta)+\epsilon} - \sqrt{\log(1/\beta)}}$, where $d_1$ is the upper bound discussed in Theorem 4.9;

15:      Add perturbation to each local model parameter $\theta_i^u = \bar{\bar{\theta}}_i^{t_1} + z_i$, where $z_i \sim \mathcal{N}(0, (n-1)\sigma^2 \mathbb{I}_d)$ is a noise from the Gaussian distribution.

16: **end for**

---

the local model $\theta_i^{t_1}$ by subtracting the accumulated gradient residual, i.e., $\theta_i^{t_1} - \sum_{t=0}^{t_1-1} r_i^t$. Instead, we provide an approximation of the gradient residual $r_i^t$, denoted by

$$\delta_i^t = \eta \sum_{j=1}^{n-1} \tilde{W}_{ij}^t \nabla F_j(\theta_j^t, \xi_j^t) - \eta \sum_{j=1}^{n} W_{ij}^t \nabla F_j(\theta_j^t, \xi_j^t) \quad (7)$$

However, this approximation $\delta_i^t$ does not fully capture the intricate dynamics and interactions among clients during training, especially the influence of client-$n$ over different rounds. To this end, we use $p_i^t$ to denote the weight of $\delta_i^t$ in the $t$-th round of the learning process. Since each client relies on the gradient information from its neighbors to update its local model, $\| \sum_{j=1}^{n} W_{ij}^t \nabla f_j(\theta_j^t, \xi_j^t) \|^2$ naturally weights the contributions of client-$i$'s neighbors' gradients in each round. Formally, $p_i^t$ is expressed as follows:

$$p_i^t = \frac{\| \sum_{j=1}^{n} W_{ij}^t \nabla F_j(\theta_j^t, \xi_j^t) \|^2}{\sum_{t=0}^{t_1-1} \| \sum_{j=1}^{n} W_{ij}^t \nabla F_j(\theta_j^t, \xi_j^t) \|^2} \qquad (8)$$

Therefore, when an unlearning request is submitted by client-$n$ in round $t_1$, each client will perform the following

operation as shown in Equation (9) to remove the influence of client-$n$ to the learning process over rounds.

$$\bar{\bar{\theta}}_i^{t_1} = \theta_i^{t_1} - \sum_{t=0}^{t_1-1} p_i^t \delta_i^t, \qquad i \in \mathcal{V}\backslash\{n\} \tag{9}$$

To achieve the indistinguishability described in Definition 4.1, we introduce random Gaussian noise to $\tilde{\theta}_i^{t_1}$ and $\bar{\bar{\theta}}_i^{t_1}$ to mask the gap between them. Specifically, Algorithm 1 outlines the perturbed retraining method, while Algorithm 2 presents our proposed decentralized unlearning algorithm that does not rely on retraining. From a global perspective, the output models of the two algorithms are indistinguishable, as will be discussed in detail in Section 4.

## 4. Theoretical Guarantee

Before delving into the unlearning performance and convergence behavior, we present some definitions and assumptions required for our theoretical analysis.

**Definition 4.1.** (($\epsilon, \beta$)-Indistinguishability (Neel et al., 2021)): Let $X$ and $Y$ be random variables over domain $\mathcal{R}$. We say that $X$ and $Y$ are ($\epsilon, \beta$)-Indistinguishable if, for every possible subset $\mathcal{S} \subseteq \mathcal{R}$, the following holds:

$$\Pr(X \in \mathcal{S}) \leq \exp(\epsilon) \cdot \Pr(Y \in \mathcal{S}) + \beta,$$

$$\Pr(Y \in \mathcal{S}) \leq \exp(\epsilon) \cdot \Pr(X \in \mathcal{S}) + \beta.$$

**Definition 4.2.** (Sensitivity (Dwork, 2006)): For a given function $q : D \to \mathbb{R}^d$, the sensitivity $\Delta$ of $q$ is

$$\Delta = \max_{D,D'} \| q(D) - q(D') \|,$$

where $D$ and $D'$ differ in a single entry.

**Definition 4.3.** (Gaussian Mechanism (Bun & Steinke, 2016)): Given random variables $X \sim \mathcal{N}(\mu_1, \sigma^2 \mathbb{I}_d)$ and $Y \sim \mathcal{N}(\mu_2, \sigma^2 \mathbb{I}_d)$ satisfying $\| \mu_1 - \mu_2 \| \leq \Delta$, then $X$ and $Y$ are ($\epsilon, \beta$)-Indistinguishable if it holds

$$\epsilon = \frac{\Delta^2}{2\sigma^2} + \frac{\Delta}{\sigma}\sqrt{2\log(1/\beta)}.$$

**Definition 4.4.** (Client-Level ($\epsilon, \beta$)-Machine Unlearning (Zhang et al., 2023)): An unlearning algorithm $\mathcal{M}_U$ satisfies ($\epsilon, \beta$)-machine unlearning with respect to the learning algorithm $\mathcal{M}_L$ if, for any possible subset of outputs $\mathcal{S} \subseteq \mathbb{R}^d$, the following holds

$$\Pr(\mathcal{M}_L(\mathcal{V}\backslash\{n\}) \in \mathcal{S}) \leq \exp(\epsilon) \cdot \Pr(\mathcal{M}_U(\Omega) \in \mathcal{S}) + \beta,$$

$$\Pr(\mathcal{M}_U(\Omega) \in \mathcal{S}) \leq \exp(\epsilon) \cdot \Pr(\mathcal{M}_L(\mathcal{V}\backslash\{n\}) \in \mathcal{S}) + \beta.$$

where $\Omega$ denotes the set of cached statistics of each client, such as gradients and intermediate model parameters.

**Assumption 4.5.** (**Lipschitzian gradient**). Loss function $f_i(\cdot)$s are with Lipschitzian gradients, i.e., For $\forall \theta, \phi \in \mathbb{R}^d$, it holds that

$$\| \nabla f_i(\theta) - \nabla f_i(\phi) \| \leq L \| \theta - \phi \|$$

**Assumption 4.6.** (**Bounded variance**). For any $\theta \in \mathbb{R}^d$, the variance of the stochastic gradient is bounded as follows:

$$\mathbb{E} \| \nabla F_i(\theta, \xi) - \nabla f_i(\theta) \|^2 \leq \sigma_1^2,$$

$$\mathbb{E} \| \nabla f_i(\theta) - \nabla f(\theta) \|^2 \leq \sigma_2^2.$$

**Assumption 4.7.** (**Symmetric double stochastic matrix**). In each round $t$, the communication matrices $\mathcal{W}^t$ and $\tilde{\mathcal{W}}^t$ are real double stochastic matrices.

**Assumption 4.8.** (**Spectral gap**). For any symmetric doubly stochastic matrices $\mathcal{W}^t$ and $\tilde{\mathcal{W}}^t$ aforementioned, we assume that $\rho_{t,1} = \max\{|\lambda_2(\mathcal{W}^t)|, |\lambda_n(\mathcal{W}^t)|\} < 1$ and $\rho_{t,2} = \max\{|\lambda_2(\tilde{\mathcal{W}}^t)|, |\lambda_{n-1}(\tilde{\mathcal{W}}^t)|\} < 1$. Specifically, we denote $\rho_1 = \max_t \rho_{t,1}$ and $\rho_2 = \max_t \rho_{t,2}$.

### 4.1. Unlearning Performance

To explore the unlearning performance of Algorithm 2, we analyze the gap between $\frac{1}{n-1}\sum_{i=1}^{n-1} \bar{\bar{\theta}}_i^{t_1}$ and $\frac{1}{n-1}\sum_{i=1}^{n-1} \tilde{\theta}_i^{t_1}$ in Theorem 4.9. We then use Definitions 4.1-4.4 to establish the statistical indistinguishability between the two algorithms, as described in Corollary 4.10.

Before obtaining the formal indistinguishability result, we first explore the difference between the average of the remaining $n-1$ models in Algorithm 2 when client-$n$ revokes data consent and the average model retrained by $n-1$ clients in Algorithm 1, as shown in Lemma B.1 of Appendix B. We then measure the weighted gradient residual approximation in Lemma B.2 of Appendix B. Based on these two lemmas, we can derive the following result for the gap between the averages $\frac{1}{n-1}\sum_{i=1}^{n-1} \bar{\bar{\theta}}_i^{t_1}$ and $\frac{1}{n-1}\sum_{i=1}^{n-1} \tilde{\theta}_i^{t_1}$.

**Theorem 4.9.** *Let*

$$D_1 = 1 - 24\eta^2 L^2 \rho_1^2 n t_1, \qquad D_2 = 1 - 36\eta^2 L^2 \rho_2^2 (n-1) t_1,$$

$$D_3 = \rho_1^2 + \rho_2^2 + \frac{1}{(n-1)^2}, \quad E_1 = \frac{(6+4\rho_1^2)\rho_1^2}{D_1},$$

$$E_2 = \frac{\rho_2^2}{D_2}, \qquad\qquad E_3 = \frac{8(1+\rho_1^2)\rho_1^2}{nD_1}.$$

*We can obtain the following result if the step size $\eta$ satisfies* $0 < \eta < \min\{\sqrt{\frac{1}{12L^2 t_1^2}}, \sqrt{\frac{1}{24L^2 \rho_1^2 n t_1}}, \sqrt{\frac{1}{36L^2 \rho_2^2 (n-1)t_1}}\}$:

$$\| \frac{1}{n-1}\sum_{i=1}^{n-1} \bar{\bar{\theta}}_i^{t_1} - \frac{1}{n-1}\sum_{i=1}^{n-1} \tilde{\theta}_i^{t_1} \|$$

$$\leq \sqrt{\begin{array}{l} I_1\sigma_1^2 + I_2\sigma_2^2 + I_3(\mathbb{E}f(\theta^0) - \mathbb{E}f(\frac{1}{n}\sum_{i=1}^{n}\theta_i^{t_1})) + \\ I_4(\mathbb{E}f(\theta^0) - \mathbb{E}f(\frac{1}{n-1}\sum_{i=1}^{n-1}\tilde{\theta}_i^{t_1})) \end{array}}$$
$$+ \sqrt{I_1'\sigma_1^2 + I_2'\sigma_2^2 + I_3'(\mathbb{E}f(\theta^0) - \mathbb{E}f(\frac{1}{n}\sum_{i=1}^{n}\theta_i^{t_1}))}$$

*where*

$$I_1 = (24\eta^4 L^2 n t_1^4 + 144\eta^5 L^3 t_1^4) \cdot (E_1 + E_2 + E_3) + 8\eta^2(5 + 2\rho_1^2)t_1^3$$

$$I_2 = 144\eta^4 L^2 n t_1^4(E_1 + 3E_2 + 2E_3) + 48\eta^2(1 + \rho_1^2)t_1^3$$

$$I_3 = 288\eta^3 L^2 n t_1^3(E_1 + E_3) + \frac{192\eta(1 + \rho_1^2)t_1^2}{n}$$

$$I_4 = 288\eta^3 L^2 n t_1^3 E_2$$

$$I_1' = 6\left(\eta^2 n + 3\eta^3 L + \frac{12\eta^4 L^2 \rho_1^2 n^2 t_1}{D_1} + \frac{72\eta^5 L^3 \rho_1^2 n t_1}{D_1}\right)D_3 t_1^2$$

$$I_2' = \left(18\eta^2 n + \frac{432\eta^4 L^2 \rho_1^2 n^2 t_1}{D_2}\right)D_3 t_1^2$$

$$I_3' = \left(36\eta n t_1 + \frac{864\eta^3 L^2 \rho_1^2 n^2 t_1^2}{D_1}\right)D_3$$

In Theorem 4.9, the terms $I_1\sigma_1^2$ and $I_1'\sigma_1^2$ reflect the impact of random sampling, the terms $I_2\sigma_2^2$ and $I_2'\sigma_2^2$ capture the local loss heterogeneity, and the remaining terms represent the decentralized training dynamics. And according to Theorem 4.9, the formal indistinguishability guarantee can be summarized as follows.

**Corollary 4.10.** *From a global perspective, our PDUDT and its early stopping variant PDUDT (ES) (described in Section 4.3) satisfies $(\epsilon, \beta)$-machine unlearning with respect to the perturbed retraining algorithm. For every possible subset $\mathcal{S} \subseteq \mathcal{R}$, it holds that*

$$\Pr\left(\frac{1}{n-1}\sum_{i=1}^{n-1}\theta_i^u \in \mathcal{S}\right) \leq \exp(\epsilon)\cdot\Pr\left(\frac{1}{n-1}\sum_{i=1}^{n-1}\check{\theta}_i^{t_1} \in \mathcal{S}\right) + \beta,$$

$$\Pr\left(\frac{1}{n-1}\sum_{i=1}^{n-1}\check{\theta}_i^{t_1} \in \mathcal{S}\right) \leq \exp(\epsilon)\cdot\Pr\left(\frac{1}{n-1}\sum_{i=1}^{n-1}\theta_i^u \in \mathcal{S}\right) + \beta.$$

### 4.2. Convergence Analysis

After each client-$i$ ($i \in \mathcal{V}\backslash\{n\}$) eliminates the influence of client-$n$ (Line 12 in Algorithm 2), we assume that it continues $T$ rounds of decentralized collaborative learning with the communication topology $\tilde{\mathcal{W}}^{t_1+t-1}$ ($t = 1, \cdots, T$). To simplify the analysis, we regard $\theta_i^u$ as the initial model of each client-$i$ ($i \in \mathcal{V}\backslash\{n\}$) in these $T$ rounds of training. Then the model update rules are as follows:

$$\hat{\theta}_i^{t+1} = \hat{\theta}_i^t - \hat{\eta}\sum_{j=1}^{n-1}\tilde{W}_{ij}^{t_1+t}\nabla F_j(\hat{\theta}_j^t, \hat{\xi}_j^t), \quad \hat{\theta}_i^0 = \theta_i^u \quad (10)$$

Considering that client-$n$ has exited the collaborative learning system, we replace $f(\cdot)$ in Equation (1) with $\tilde{f}(\cdot)$, which is defined as:

$$\tilde{f}(\theta) := \frac{1}{n-1}\sum_{i=1}^{n-1}f_i(\theta), \quad \theta \in \mathbb{R}^d \quad (11)$$

As a result, after eliminating the client-$n$ influence through Algorithm 2, the convergence of decentralized learning in the subsequent $T$ rounds is characterized by Theorem 4.11.

**Theorem 4.11.** *Let*

$$D_4 = \frac{1}{2} - \frac{16\hat{\eta}^2 L^2 \rho_2^2(n-1)T}{1 - 32\hat{\eta}^2 L^2 \rho_2^2(n-1)T}, \quad D_5 = \frac{1}{2} - \frac{\hat{\eta}L}{2}$$

$$D_6 = 1 - 32\hat{\eta}^2 L^2 \rho_2^2(n-1)T.$$

*If the step size satisfies $\hat{\eta} < \sqrt{\frac{1}{32L^2\rho_2^2(n-1)T}}$, it holds for the subsequent $T$ rounds of training:*

$$D_4 \cdot \frac{1}{T}\sum_{t=0}^{T-1}\mathbb{E}\parallel\nabla\tilde{f}(\frac{1}{n-1}\sum_{i=1}^{n-1}\hat{\theta}_i^t)\parallel^2 +$$

$$D_5\frac{1}{T}\sum_{t=0}^{T-1}\mathbb{E}\parallel\frac{1}{n-1}\sum_{i=1}^{n-1}\nabla\tilde{f}(\hat{\theta}_i^t)\parallel^2$$

$$\leq \frac{3L^2}{2D_6}\cdot\frac{1}{n-1}\sum_{i=1}^{n-1}\mathbb{E}\parallel\theta_i^u - \frac{1}{n-1}\sum_{i=1}^{n-1}\theta_i^u\parallel^2 + \frac{\hat{\eta}L\sigma_1^2}{2(n-1)}$$

$$+ \frac{\tilde{f}(\frac{1}{n-1}\sum_{i=1}^{n-1}\theta_i^u) - \tilde{f}^*}{\hat{\eta}T} + \frac{2\hat{\eta}^2 L^2\rho_2^2\sigma_1^2(n-1)T}{D_6}$$

$$+ \frac{16\hat{\eta}^2 L^2\rho_2^2\sigma_2^2(n-1)T}{D_6} + \frac{16\hat{\eta}^2 L^2\rho_2^2\sigma_2^2 T}{(n-1)D_6}$$

We choose an appropriate step size $\hat{\eta}$ in Theorem 4.9 to derive the following result.

**Corollary 4.12.** *If the step size satisfies $\hat{\eta} = \frac{n-1}{T}$, and the number of training round further satisfies $T \geq \max\{\frac{64(1-C)L^2\rho_2^2(n-1)^3}{1-2C}, (n-1)L\}$ with the constant $C \in (0, \frac{1}{2})$, the following holds for the subsequent $T$ rounds:*

$$C \cdot \frac{1}{T}\sum_{t=0}^{T-1}\mathbb{E}\parallel\nabla\tilde{f}(\frac{1}{n-1}\sum_{i=1}^{n-1}\hat{\theta}_i^t)\parallel^2$$

$$\leq \frac{L\sigma_1^2}{2T} + \frac{\tilde{f}(\frac{1}{n-1}\sum_{i=1}^{n-1}\theta_i^u) - \tilde{f}^*}{n-1} + \frac{(1-2C)\sigma_2^2}{2(n-1)^2} + \frac{(1-2C)\sigma_1^2}{16}$$

$$+ \frac{(1-2C)\sigma_2^2}{2} + \frac{3(1-C)L^2}{n-1}\sum_{i=1}^{n-1}\mathbb{E}\parallel\theta_i^u - \frac{1}{n-1}\sum_{i=1}^{n-1}\theta_i^u\parallel^2$$

In Corollary 4.12, the bound sharply decreases as the number of clients $n-1$ and training rounds $T$ increase. Specifically, it indicates a rate of $\mathcal{O}(\frac{1}{T})$, converging to the scaled difference among the unlearning models from all $n-1$ clients.

### 4.3. Discussion

In this part, we provide a discussion of our decentralized unlearning algorithm PDUDT regarding privacy considerations, unlearning time, space overhead, and early stopping benefits.

**Privacy considerations.** Some works have shown that transmitting the raw gradient can also lead to privacy information leakage. Differential privacy technology, as the most commonly used privacy protection method for distributed learning, provides a solution to this concern by adding noise to perturb the gradient. For these decentralized learning frameworks based on differential privacy, the perturbed information of client-$n$ can be regarded as its contribution to the system. As a result, our proposed mechanism can still be used to eliminate the noisy impact of client-$n$.

**Unlearning time.** The proposed algorithm involves only simple operations (Lines 7-12 in Algorithm 2) to achieve unlearning. Let $t_1$ denote the training round when client-$n$ submits an unlearning request. For each client, the time complexity to remove the contribution of client-$n$ is $\mathcal{O}(t_1)$. Specifically, subtracting the weighted gradient residual approximation requires $\mathcal{O}(t_1)$ time, while computing the distance $d_1$ and sampling noise from a Gaussian distribution each require $\mathcal{O}(1)$ time. Furthermore, the unlearning operations are only performed on each client side, so the required communication overhead is 0.

**Space overhead.** The proposed decentralized unlearning algorithm requires each client-$i$ to save the gradients of its neighbors. Let $N_{\max} = \max_{i,t} |N_i^t|$ denote the maximum number of neighbors over rounds $t = 0, \ldots, t_1 - 1$. For each client, it costs $\mathcal{O}(t_1 N_{\max} d)$ in memory space to perform the unlearning operations. Although $N_{\max} \leq n$ always holds, especially in the scenarios like social networks, the required memory increases linearly as the number of neighbors grows. To further reduce memory usage, each client may consider an early stopping strategy based on its resource constraints and model performance.

**Early stopping benefits.** For neural networks in an over-parameterization regime, the NTK theory (Lee et al., 2019) suggests that gradient-based methods converge exponentially to zero training error, with minimal variation in the model's parameters (Zhang et al., 2023). Therefore, each client needs to store only the gradients of its neighbors for the first $t_{1,i} \leq t_1$ rounds to save memory. Under this setup, we can redefine Lines 8-10 in Algorithm 2: each client-$i$ computes the approximations $\{\delta_i^t\}_{t=0}^{t_{1,i}-1}$ of the gradient residuals using Equation (7). Then, it calculates the weight $p_i^{'t}$ for every approximation $\delta_i^t$ ($t = 0, \cdots, t_{1,i} - 1$) as

$$p_i^{'t} = \frac{\| \sum_{j=1}^n W_{ij}^t \nabla f_j(\theta_j^t, \xi_j^t) \|^2}{\sum_{t=0}^{t_{1,i}-1} \| \sum_{j=1}^n W_{ij}^t \nabla f_j(\theta_j^t, \xi_j^t) \|^2}$$

and subtract a weighted sum of $\delta_i^t$ from $\theta_i^{t_1}$ to obtain $\bar{\bar{\theta}}_i^{t_1}$:

$$\bar{\bar{\theta}}_i^{t_1} = \theta_i^{t_1} - \sum_{t=0}^{t_{1,i}-1} p_i^{'t} \delta_i^t, \quad i \in \mathcal{V} \backslash \{n\}.$$

Furthermore, we can prove that the weighted gradient residual approximation with early stopping is also bounded by the right-hand side of the inequality in Lemma B.2, as detailed in Appendix D.

## 5. Experiments

In this section, we evaluate PDUDT from many aspects, such as its statistical indistinguishability from the perturbed retraining algorithm, as well as its efficiency and effectiveness of unlearning.

### 5.1. Experimental Setup

According to the complexity of the learning tasks, we train the CNN model for MNIST (Lecun et al., 1998) and Fashion-MNIST (Xiao et al., 2017) datasets, and the ResNet-18 model (He et al., 2016) for CIFAR-10 (Krizhevsky & Hinton, 2009) and SVHN (Netzer et al., 2011) datasets. To show the advantages of our proposed PDUDT algorithm, we compare it with some baseline methods, including *Origin* (Lian et al., 2017), *Retrain*, *FATS-Unl* (Tao et al., 2024), *FedRecovery* (Zhang et al., 2023) and *HDUS* (Ye et al., 2024). Multiple metrics are used to evaluate the performance of the proposed decentralized unlearning algorithm, including accuracy, unlearning time, communication overhead, and attack success rate. More details can be found in Appendix I.

In our experiments, we work with total $n = 10$ clients. Specifically, in each round $t$, whether there is a connection between any two clients is randomly generated. Then, in order to ensure that the communication situation can be modeled as a doubly stochastic matrix, we use the Metropolis-Hastings method (Awan et al., 2006) to generate the communication weights among clients. Each client trains with a batch size of 256 for 1 epoch per round, with a step size of $0.001$. The unlearning request from client-$n$ is set to occur at round $t_1 = 100$. To save storage space, each client can apply an early stopping strategy by retaining its neighbors' information only for the first 80 rounds. After performing the unlearning operations, the remaining $n - 1$ clients continue training collaboratively for an additional 200 rounds. We conduct 100 membership inference attacks, presenting both the average performance and standard deviation.

Our experiments are conducted using PyTorch 2.5.1, Python 3.12, and Cuda 12.1. The experiments run on a cloud server equipped with an Intel(R) Xeon(R) Platinum 8358P CPU and 10 RTX 3090 GPUs, operating on Ubuntu 22.04.

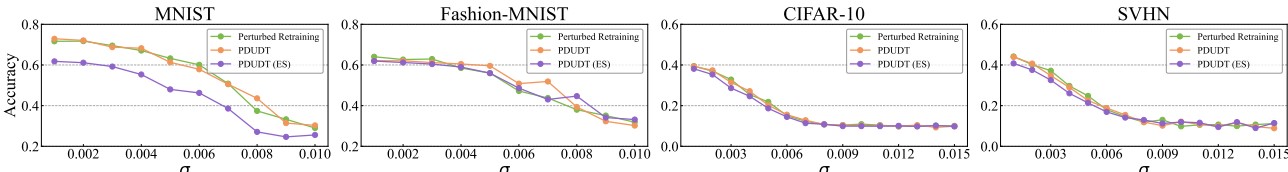

*Figure 1.* The accuracy of unlearned models using PDUDT, PDUDT (ES), and perturbed retrained models.

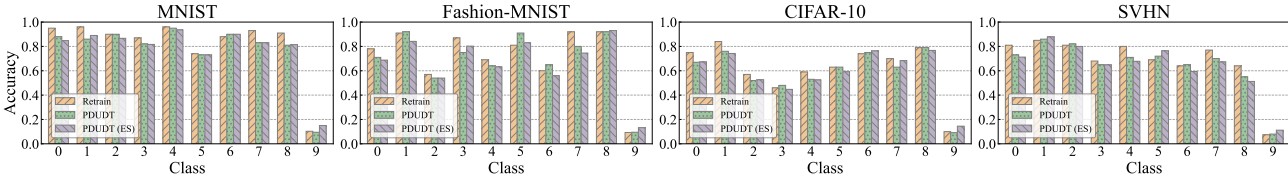

*Figure 2.* The accuracy on each class using PDUDT, PDUDT (ES), and perturbed retrained models.

*Table 2.* Comparison of the unlearning time across different unlearning methods.

| Method | Unlearning time (s) | | | |
|---|---|---|---|---|
| | MNIST | Fashion-MNIST | CIFAR-10 | SVHN |
| Retrain | 1322.2 | 1299.8 | 1696.2 | 2747.8 |
| FATS-Unl | 573.6 | 573.4 | 884.0 | 1233.2 |
| FedRecovery | 3.5 | 3.9 | 8.1 | 8.5 |
| HDUS | 19.3 | 19.9 | 24.1 | 38.0 |
| **PDUDT** | 3.0 | 3.3 | 9.6 | 9.8 |
| **PDUDT (ES)** | 2.3 | 2.6 | 7.5 | 7.8 |

*Table 3.* Comparison of the attack precision of MIA across different unlearning methods.

| Method | Attack precision (%) | | | |
|---|---|---|---|---|
| | MNIST | Fashion-MNIST | CIFAR-10 | SVHN |
| Origin | $65.3 \pm 0.9$ | $67.1 \pm 1.9$ | $65.3 \pm 0.7$ | $68.1 \pm 0.5$ |
| Retrain | $49.4 \pm 1.3$ | $49.2 \pm 2.2$ | $51.0 \pm 0.6$ | $49.5 \pm 0.7$ |
| FATS-Unl | $51.3 \pm 1.2$ | $49.6 \pm 4.3$ | $53.1 \pm 1.2$ | $51.8 \pm 1.6$ |
| FedRecovery | $50.1 \pm 0.7$ | $48.4 \pm 0.9$ | $52.5 \pm 0.6$ | $51.1 \pm 0.3$ |
| HDUS | $50.2 \pm 1.1$ | $51.2 \pm 0.7$ | $50.6 \pm 0.3$ | $51.2 \pm 1.3$ |
| **PDUDT** | $49.9 \pm 0.6$ | $49.0 \pm 1.3$ | $50.1 \pm 1.6$ | $51.3 \pm 0.3$ |
| **PDUDT (ES)** | $50.8 \pm 0.6$ | $50.2 \pm 0.9$ | $51.1 \pm 1.3$ | $51.5 \pm 1.4$ |

## 5.2. Experimental Results

**1) Statistical indistinguishability:** To examine the effect of different noise scales on the statistical indistinguishability between unlearned models and perturbed retrained models, we vary the values of $\sigma$ from 0.001 to 0.015. The performance is evaluated in terms of the average model accuracy. Specifically, we start decentralized learning from pre-trained models to control the injected noise and ensure good performance. Figure 1 illustrates the statistical indistinguishability

of our PDUDT algorithm from the Perturbed Retraining method under varying noise scales $\sigma$. Across all datasets, PDUDT achieves comparable accuracy to the Perturbed Retraining method under all noise conditions, demonstrating its ability to maintain statistical indistinguishability effectively. Similarly, PDUDT (ES), the space-saving version of PDUDT, also shows comparable performance, though with slight degradation in MNIST dataset. Overall, the results confirm that both PDUDT and PDUDT (ES) can maintain statistical indistinguishability with perturbed retrained models across all noise conditions.

**2) The Effectiveness of Unlearning:** To evaluate the effectiveness of PDUDT and its space-saving version PDUDT (ES), we record the average accuracy on each class. In this experiment, data from class 9 is exclusively owned by the client who requests unlearning. Therefore, the unlearning performance can be assessed based on the accuracy of this class. From Figure 2, it can be observed that both PDUDT and PDUDT (ES) maintain high performance on Classes 0–8, while their accuracy drops significantly on Class 9, exhibiting behavior similar to the retraining method. This demonstrates that PDUDT and PDUDT (ES) effectively enable the entire system to forget what it has learned from the client initiating the unlearning request.

**3) The Efficiency of Unlearning:** To verify the efficiency of PDUDT and PDUDT (ES), we evaluate the time required for unlearning operations. In Table 2, our PDUDT and its space-saving version PDUDT (ES) show impressive performance in both unlearning time. Specifically, they reduce unlearning time by over 99% compared to the Retrain method. The time consumption of FedRecovery, HDUS, PDUDT and PDUDT (ES) primarily arises from computing gradient residuals or

the stored distilled seed models ensemble. In contrast, the time consumption for Retrain and FATS-Unl is mainly due to parameter training within the networks.

**4) The Performance of MIA on Unlearned Models:** In this experiment, we conduct 100 membership inference attacks, presenting both the average performance and standard deviation across different unlearning methods. As shown in Table 3, the MIA achieves high attack precision on the original models, indicating that the attacker can successfully determine whether the target client's data was used during training. In contrast, after unlearning, its attack precision drops to approximately 50%, demonstrating that the proposed unlearning methods successfully remove the impact of the target client. Notably, the performance of PDUDT and PDUDT (ES) is comparable to that of Retrain and FATS-Unl, highlighting that our PDUDT and PDUDT (ES) methods achieve similar unlearning effectiveness without relying on retraining.

## 6. Conclusion

In this paper, we propose the first provable decentralized unlearning algorithm PDUDT under dynamic topologies. Theoretically, we derive its statistical indistinguishability and the convergence of its subsequent learning process. We conduct extensive experiments to verify the performance superiority of the proposed unlearning algorithm.

Our work inspires future research on provable decentralized unlearning. It paves the way for investigating adaptive mechanisms that enhance unlearning efficiency under dynamic topologies and for extending the unlearning framework to broader decentralized paradigms.

## Acknowledgements

This work was supported in part by Joint Key Funds of National Natural Science Foundation of China under Grant U23A20302 and U24B20149, in part by the National Natural Science Foundation of China under Grant 62202273 and 62302247, in part by the National Key Research and Development Program of China under Grant No. 2022YFF0712100, in part by the Postdoctoral Fellowship Program of CPSF under Grant GZC20231460, in part by the China Postdoctoral Science Foundation under Grant 2024M761806.

## Impact Statement

This paper presents work whose goal is to advance the field of Machine Learning. There are many potential societal consequences of our work, none which we feel must be specifically highlighted here.

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

## A. Notation Table

*Table 4.* Notations and descriptions.

| Notations | Descriptions |
|---|---|
| $t_1$ | The time when client-$n$ issues an unlearning request |
| $\eta$ | The step size for Algorithms 1-2 |
| $\sigma$ | The noise scale in Algorithms 1-2 |
| $\epsilon$ | The privacy budget of indistinguishability |
| $\beta$ | The confidence parameter of indistinguishability |
| $\tilde{\theta}_i^t$ | The local model of client-$i$ in round $t$ ($t = 1, \cdots, t_1$) in Algorithm 1 |
| $\theta_i^t$ | The local model of client-$i$ in round $t$ ($t = 1, \cdots, t_1$) in Algorithm 2 |
| $\check{\theta}_i^{t_1}$ | The perturbed retrained model of client-$i$ in Algorithm 1 |
| $r_i^t$ | The gradient residual in round $t$ related to client-$i$'s neighbors' information |
| $\delta_i^t$ | The approximation of the gradient residual $r_i^t$ in round $t$, which is computed by client-$i$ |
| $p_i^t$ | The weight of $\delta_i^t$, which measures the contributions of client-$i$'s neighbors' gradients in round $t$ |
| $\bar{\bar{\theta}}_i^{t_1}$ | The local model of client-$i$ after removing the influence of client-$n$ based on Equation (9) |
| $\theta_i^u$ | The unlearned model of client-$i$ in Algorithm 2 |
| $\nabla F_i(\theta_i^t, \xi_i^t)$ | The local gradient of client-$i$ related to model $\theta_i^t$ and sample $\xi_i^t$ in round $t$ |

## B. Some important theoretical results

Before obtaining the formal indistinguishability result, we first explore the difference between the average of the remaining $n - 1$ models in Algorithm 2 when client-$n$ revokes data consent and the average model retrained by $n - 1$ clients in Algorithm 1, as shown in Lemma B.1. We then measure the weighted gradient residual approximation in Lemma B.2.

**Lemma B.1.** *Let*

$$D_1 = 1 - 24\eta^2 L^2 \rho_1^2 n t_1, \qquad D_2 = 1 - 36\eta^2 L^2 \rho_2^2 (n-1) t_1.$$

*If* $0 < \eta < \min\{\sqrt{\frac{1}{12L^2 t_1^2}}, \sqrt{\frac{1}{24L^2 \rho_1^2 n t_1}}, \sqrt{\frac{1}{36L^2 \rho_2^2 (n-1) t_1}}\}$, *it holds that*

$$\| \frac{1}{n-1} \sum_{i=1}^{n-1} \theta_i^{t_1} - \frac{1}{n-1} \sum_{i=1}^{n-1} \tilde{\theta}_i^{t_1} \|$$

$$\leq \sqrt{ \begin{aligned} & (24\eta^4 L^2 \sigma_1^2 n t_1^4 + 144\eta^5 L^3 \sigma_1^2 t_1^4) \cdot (\frac{(6+4\rho_1^2)\rho_1^2}{D_1} + \frac{\rho_2^2}{D_2} + \frac{8(1+\rho_1^2)\rho_1^2}{nD_1}) + 8\eta^2(5 + 2\rho_1^2)\sigma_1^2 t_1^3 + 48\eta^2(1 + \rho_1^2)\sigma_2^2 t_1^3 + \\ & 144\eta^4 L^2 \sigma_2^2 t_1^4 \cdot (\frac{(6+4\rho_1^2)\rho_1^2 n}{D_1} + \frac{3\rho_2^2(n-1)}{D_2} + \frac{16(1+\rho_1^2)\rho_1^2}{D_1}) + (\frac{2304\eta^3 L^2 \rho_1^2(1+\rho_1^2) t_1^3}{D_1} + \frac{576\eta^3 L^2 \rho_1^2(3+2\rho_1^2) n t_1^3}{D_1} + \\ & \frac{192\eta(1+\rho_1^2) t_1^2}{n}) \cdot (\mathbb{E}f(\theta^0) - \mathbb{E}f(\frac{1}{n}\sum_{i=1}^{n} \theta_i^{t_1})) + \frac{288\eta^3 L^2 \rho_2^2(n-1) t_1^3}{D_2} \cdot (\mathbb{E}f(\theta^0) - \mathbb{E}f(\frac{1}{n-1}\sum_{i=1}^{n-1} \tilde{\theta}_i^{t_1})) \end{aligned} }$$

**Lemma B.2.** *Let*

$$D_1 = 1 - 24\eta^2 L^2 \rho_1^2 n t_1, \qquad D_2 = 1 - 36\eta^2 L^2 \rho_2^2 (n-1) t_1$$

$$D_3 = \rho_1^2 + \rho_2^2 + \frac{1}{(n-1)^2}.$$

*If $0 < \eta < \min\{\sqrt{\frac{1}{12L^2 t_1^2}}, \sqrt{\frac{1}{24L^2 \rho_1^2 n t_1}}, \sqrt{\frac{1}{36L^2 \rho_2^2 (n-1) t_1}}\}$, the norm of the weighted gradient residual approximation (with early stopping, discussed in Section 4.3) holds that*

$$\| \frac{1}{n-1} \sum_{i=1}^{n-1} \sum_{t=0}^{t_1-1} p_i^t \delta_i^t \|$$

$$(\textit{Early stopping: } \| \frac{1}{n-1} \sum_{i=1}^{n-1} \sum_{t=0}^{t_{1,i}-1} p_i^{'t} \delta_i^t \|)$$

$$\leq \sqrt{ \begin{array}{l} 6\big(\eta^2 n + 3\eta^3 L + \frac{12\eta^4 L^2 \rho_1^2 n^2 t_1}{D_1} + \frac{72\eta^5 L^3 \rho_1^2 n t_1}{D_1}\big) D_3 \sigma_1^2 t_1^2 + \big(18\eta^2 n + \frac{432\eta^4 L^2 \rho_1^2 n^2 t_1}{D_2}\big) D_3 \sigma_2^2 t_1^2 + \\ \big(36\eta n t_1 + \frac{864\eta^3 L^2 \rho_1^2 n^2 t_1^2}{D_1}\big) D_3 (\mathbb{E} f(\theta^0) - \mathbb{E} f(\frac{1}{n}\sum_{i=1}^{n} \theta_i^{t_1})) \end{array} }$$

# C. Proof of Lemma B.1

Based on Equation (5), (9), (4) and (7), we can directly obtain Equation (12)-(15):

$$\frac{1}{n-1} \sum_{i=1}^{n-1} \tilde{\theta}_i^{t_1} = \tilde{\theta}_i^0 - \frac{\eta}{n-1} \sum_{t=0}^{t_1-1} \sum_{j=1}^{n-1} \nabla F_j(\tilde{\theta}_j^t, \tilde{\xi}_j^t) \tag{12}$$

$$\frac{1}{n-1} \sum_{i=1}^{n-1} \bar{\bar{\theta}}_i^{t_1} = \theta_i^0 - \frac{\eta}{n-1} \sum_{t=0}^{t_1-1} \big( \sum_{i=1}^{n} \sum_{j=1}^{n} W_{ij}^t \nabla F_j(\theta_j^t, \xi_j^t) - \sum_{j=1}^{n} W_{nj}^t \nabla F_j(\theta_j^t, \xi_j^t) - \sum_{i=1}^{n} p_i^t \delta_i^t \big)$$

$$= \theta_i^0 - \frac{\eta}{n-1} \sum_{t=0}^{t_1-1} \sum_{j=1}^{n} \nabla F_j(\theta_j^t, \xi_j^t) + \frac{\eta}{n-1} \sum_{t=0}^{t_1-1} \big( \sum_{j=1}^{n} W_{nj}^t \nabla F_j(\theta_j^t, \xi_j^t) - \sum_{i=1}^{n} p_i^t \delta_i^t \big) \tag{13}$$

$$\frac{1}{n-1} \sum_{i=1}^{n-1} \theta_i^{t_1} = \theta_i^0 - \frac{\eta}{n-1} \sum_{t=0}^{t_1-1} \sum_{j=1}^{n} \nabla F_j(\theta_j^t, \xi_j^t) + \frac{\eta}{n-1} \sum_{t=0}^{t_1-1} \sum_{j=1}^{n} W_{nj}^t \nabla F_j(\theta_j^t, \xi_j^t) \tag{14}$$

$$\frac{1}{n-1} \sum_{i=1}^{n-1} \delta_i^t = \frac{\eta}{n-1} \sum_{i=1}^{n-1} \sum_{j=1}^{n-1} \tilde{W}_{ij}^t \nabla F_j(\theta_j^t, \xi_j^t) - \frac{\eta}{n-1} \sum_{i=1}^{n} \sum_{j=1}^{n} W_{ij}^t \nabla F_j(\theta_j^t, \xi_j^t) + \frac{\eta}{n-1} \sum_{j=1}^{n} W_{nj}^t \nabla F_j(\theta_j^t, \xi_j^t)$$

$$= \frac{\eta}{n-1} \big( \sum_{j=1}^{n} W_{nj}^t \nabla F_j(\theta_j^t, \xi_j^t) - \nabla F_n(\theta_n^t, \xi_n^t) \big) \tag{15}$$

Substitute Equation (15) into Equation (13) to get

$$\frac{1}{n-1} \sum_{i=1}^{n-1} \bar{\bar{\theta}}_i^{t_1} = \theta_i^0 - \frac{\eta}{n-1} \sum_{t=0}^{t_1-1} \sum_{j=1}^{n} \nabla F_j(\theta_j^t, \xi_j^t) + \frac{\eta}{n-1} \sum_{t=0}^{t_1-1} \big( \nabla F_n(\theta_n^t, \xi_n^t) + \sum_{i=1}^{n} (1 - p_i^t) \delta_i^t \big) \tag{16}$$

Consider the gap

$$\frac{1}{n-1} \sum_{i=1}^{n-1} \theta_i^{t_1} - \frac{1}{n-1} \sum_{i=1}^{n-1} \tilde{\theta}_i^{t_1}$$

$$= -\frac{\eta}{n-1} \sum_{t=0}^{t_1-1} \sum_{j=1}^{n} \nabla F_j(\theta_j^t, \xi_j^t) + \frac{\eta}{n-1} \sum_{t=0}^{t_1-1} \sum_{j=1}^{n-1} \nabla F_j(\tilde{\theta}_j^t, \tilde{\xi}_j^t) + \frac{\eta}{n-1} \sum_{t=0}^{t_1-1} \sum_{j=1}^{n} W_{nj}^t \nabla F_j(\theta_j^t, \xi_j^t)$$

$$= \frac{\eta}{n-1} \sum_{t=0}^{t_1-1} \sum_{j=1}^{n-1} \big( \nabla F_j(\tilde{\theta}_j^t, \tilde{\xi}_j^t) - \nabla F_j(\theta_j^t, \xi_j^t) \big) + \frac{\eta}{n-1} \sum_{t=0}^{t_1-1} \big( \sum_{j=1}^{n} W_{nj}^t \nabla F_j(\theta_j^t, \xi_j^t) - \nabla F_n(\theta_n^t, \xi_n^t) \big) \tag{17}$$

Then we have

$$\mathbb{E} \parallel \frac{1}{n-1} \sum_{i=1}^{n-1} \theta_i^{t_1} - \frac{1}{n-1} \sum_{i=1}^{n-1} \tilde{\theta}_i^{t_1} \parallel^2$$

$$\leq \frac{2\eta^2 t_1}{n-1} \sum_{t=0}^{t_1-1} \sum_{j=1}^{n-1} \mathbb{E} \parallel \nabla F_j(\tilde{\theta}_j^t, \tilde{\xi}_j^t) - \nabla f_j(\tilde{\theta}_j^t) + \nabla f_j(\tilde{\theta}_j^t) - \nabla f_j(\frac{1}{n-1} \sum_{j=1}^{n-1} \tilde{\theta}_j^t) + \nabla f_j(\frac{1}{n-1} \sum_{j=1}^{n-1} \tilde{\theta}_j^t) -$$

$$\nabla f_j(\frac{1}{n-1} \sum_{j=1}^{n-1} \theta_j^t) + \nabla f_j(\frac{1}{n-1} \sum_{j=1}^{n-1} \theta_j^t) - \nabla f_j(\frac{1}{n} \sum_{j=1}^{n-1} \theta_j^t) + \nabla f_j(\frac{1}{n} \sum_{j=1}^{n-1} \theta_j^t) - \nabla f_j(\theta_j^t) + \nabla f_j(\theta_j^t) - \nabla F_j(\theta_j^t, \xi_j^t) \parallel^2$$

$$+ \frac{2\eta^2 t_1}{(n-1)^2} \sum_{t=0}^{t_1-1} \mathbb{E} \parallel \sum_{j=1}^{n} W_{nj}^t \nabla F_j(\theta_j^t, \xi_j^t) - \frac{1}{n} \sum_{j=1}^{n} \nabla F_j(\theta_j^t, \xi_j^t) + \frac{1}{n} \sum_{j=1}^{n} \nabla F_j(\theta_j^t, \xi_j^t) - \nabla F_n(\theta_n^t, \xi_n^t) \parallel^2$$

$$= 24\eta^2 t_1^2 \sigma_1^2 + \frac{12\eta^2 L^2 t_1}{n-1} \sum_{t=0}^{t_1-1} \sum_{j=1}^{n-1} \mathbb{E} \parallel \tilde{\theta}_j^t - \frac{1}{n-1} \sum_{j=1}^{n-1} \tilde{\theta}_j^t \parallel^2 + \frac{12\eta^2 L^2 t_1}{n-1} \sum_{t=0}^{t_1-1} \sum_{j=1}^{n-1} \mathbb{E} \parallel \frac{1}{n-1} \sum_{j=1}^{n-1} \tilde{\theta}_j^t - \frac{1}{n-1} \sum_{j=1}^{n-1} \theta_j^t \parallel^2 +$$

$$\frac{12\eta^2 L^2 t_1}{(n-1)^2} \sum_{t=0}^{t_1-1} \mathbb{E} \parallel \theta_n^t - \frac{1}{n} \sum_{i=1}^{n} \theta_i^t \parallel^2 + \frac{12\eta^2 L^2 t_1}{n-1} \sum_{t=0}^{t_1-1} \sum_{i=1}^{n-1} \mathbb{E} \parallel \theta_i^t - \frac{1}{n} \sum_{i=1}^{n} \theta_i^t \parallel^2 +$$

$$\frac{4\eta^2 t_1}{(n-1)^2} \sum_{t=0}^{t_1-1} \mathbb{E} \parallel (e_n^T W^t - \frac{1_n^T}{n}) G_{(n)}^t \parallel^2 + \frac{4\eta^2 t_1}{(n-1)^2} \sum_{t=0}^{t_1-1} \mathbb{E} \parallel (\frac{1_n^T}{n} - e_n^T) G_{(n)}^t \parallel^2$$

$$\leq 24\eta^2 t_1^2 \sigma_1^2 + \frac{12\eta^2 L^2 t_1}{n-1} \sum_{t=0}^{t_1-1} \sum_{i=1}^{n-1} \mathbb{E} \parallel \tilde{\theta}_i^t - \frac{1}{n-1} \sum_{i=1}^{n-1} \tilde{\theta}_i^t \parallel^2 + \frac{12\eta^2 L^2 t_1}{n-1} \sum_{t=0}^{t_1-1} \sum_{i=1}^{n-1} \mathbb{E} \parallel \frac{1}{n-1} \sum_{i=1}^{n-1} \tilde{\theta}_i^t - \frac{1}{n-1} \sum_{i=1}^{n-1} \theta_i^t \parallel^2 +$$

$$\frac{12\eta^2 L^2 t_1}{n-1} \sum_{t=0}^{t_1-1} \sum_{i=1}^{n} \mathbb{E} \parallel \theta_i^t - \frac{1}{n} \sum_{i=1}^{n} \theta_i^t \parallel^2 + \frac{4\eta^2 (1+\rho_1^2) t_1}{(n-1)^2} \sum_{t=0}^{t_1-1} \mathbb{E} \parallel G_{(n)}^t - H_{(n)}^t + H_{(n)}^t \parallel^2 \tag{18}$$

Bound $\mathbb{E} \parallel G_{(n)}^t - H_{(n)}^t + H_{(n)}^t \parallel^2$:

$$\mathbb{E} \parallel G_{(n)}^t - H_{(n)}^t + H_{(n)}^t \parallel^2$$

$$\leq 2 \sum_{i=1}^{n} \mathbb{E} \parallel \nabla F_i(\theta_i^t, \xi_i^t) - \nabla f_i(\theta_i^t) \parallel^2 +$$

$$2 \sum_{i=1}^{n} \mathbb{E} \parallel \nabla f_i(\theta_i^t) - \nabla f_i(\frac{1}{n} \sum_{i=1}^{n} \theta_i^t) + \nabla f_i(\frac{1}{n} \sum_{i=1}^{n} \theta_i^t) - \nabla f(\frac{1}{n} \sum_{i=1}^{n} \theta_i^t) + \nabla f(\frac{1}{n} \sum_{i=1}^{n} \theta_i^t) \parallel^2$$

$$\leq 2n\sigma_1^2 + 6L^2 \sum_{i=1}^{n} \mathbb{E} \parallel \theta_i^t - \frac{1}{n} \sum_{i=1}^{n} \theta_i^t \parallel^2 + 6n\sigma_2^2 + 6n\mathbb{E} \parallel \nabla f(\frac{1}{n} \sum_{i=1}^{n} \theta_i^t) \parallel^2 \tag{19}$$

Bound $\mathbb{E} \parallel \tilde{\theta}_i^t - \frac{1}{n-1} \sum_{i=1}^{n-1} \tilde{\theta}_i^t \parallel^2$:

$$\mathbb{E} \parallel \tilde{\theta}_i^t - \frac{1}{n-1} \sum_{i=1}^{n-1} \tilde{\theta}_i^t \parallel^2$$

$$= \mathbb{E} \parallel \eta \sum_{l=0}^{t-1} \Big( \frac{1}{n-1} \sum_{j=1}^{n-1} \nabla F_j(\tilde{\theta}_j^l, \tilde{\xi}_j^l) - \sum_{j=1}^{n-1} \tilde{W}_{ij}^l \nabla F_j(\tilde{\theta}_j^l, \tilde{\xi}_j^l) \Big) \parallel^2$$

$$= \eta^2 \mathbb{E} \parallel \sum_{l=0}^{t-1} (\frac{1_{n-1}^T}{n-1} - \tilde{e}_i^T \tilde{W}^l)(\tilde{G}_{(n-1)}^l - \tilde{H}_{(n-1)}^l + \tilde{H}_{(n-1)}^l) \parallel^2$$

$$\leq 2\eta^2 \Big( \sum_{l=0}^{t-1} \mathbb{E} \parallel (\frac{1_{n-1}^T}{n-1} - \tilde{e}_i^T \tilde{W}^l)(\tilde{G}_{(n-1)}^l - \tilde{H}_{(n-1)}^l) \parallel^2 + \sum_{l=0}^{t-1} \mathbb{E} \parallel (\frac{1_{n-1}^T}{n-1} - \tilde{e}_i^T \tilde{W}^l) \tilde{H}_{(n-1)}^l \parallel^2 +$$

$$\sum_{l' \neq l}^{t-1} \mathbb{E} \langle (\frac{1_{n-1}^T}{n-1} - \tilde{e}_i^T \tilde{W}^l) \tilde{H}_{(n-1)}^l, (\frac{1_{n-1}^T}{n-1} - \tilde{e}_i^T \tilde{W}^{l'}) \tilde{H}_{(n-1)}^{l'} \rangle)$$

$$\leq 2\eta^2 \rho_2^2 \sigma_1^2 (n-1)t + 2\eta^2 \big(\rho_2^2 \sum_{l=0}^{t-1} \mathbb{E} \parallel \tilde{H}_{(n-1)}^l \parallel^2 + \sum_{l' \neq l}^{t-1} \mathbb{E} \langle (\frac{1_{n-1}^T}{n-1} - \tilde{e}_i^T \tilde{W}^l) \tilde{H}_{(n-1)}^l, (\frac{1_{n-1}^T}{n-1} - \tilde{e}_i^T \tilde{W}^{l'}) \tilde{H}_{(n-1)}^{l'} \rangle) \quad (20)$$

And

$$\sum_{l=0}^{t-1} \mathbb{E} \parallel \tilde{H}_{(n-1)}^l \parallel^2$$

$$= \sum_{l=0}^{t-1} \sum_{i=1}^{n-1} \mathbb{E} \parallel \nabla f_i(\tilde{\theta}_i^l) - \nabla f_i(\frac{1}{n-1} \sum_{i=1}^{n-1} \tilde{\theta}_i^l) + \nabla f_i(\frac{1}{n-1} \sum_{i=1}^{n-1} \tilde{\theta}_i^l) - \nabla f(\frac{1}{n-1} \sum_{i=1}^{n-1} \tilde{\theta}_i^l) + \nabla f(\frac{1}{n-1} \sum_{i=1}^{n-1} \tilde{\theta}_i^l) \parallel^2$$

$$\leq 3L^2 \sum_{l=0}^{t-1} \sum_{i=1}^{n-1} \mathbb{E} \parallel \tilde{\theta}_i^l - \frac{1}{n-1} \sum_{i=1}^{n-1} \tilde{\theta}_i^l \parallel^2 + 3\sigma_2^2(n-1)t + 3(n-1) \sum_{l=0}^{t-1} \mathbb{E} \parallel \nabla f(\frac{1}{n-1} \sum_{i=1}^{n-1} \tilde{\theta}_i^l) \parallel^2 \quad (21)$$

$$\sum_{l' \neq l}^{t-1} \mathbb{E} \langle (\frac{1_{n-1}^T}{n-1} - \tilde{e}_i^T \tilde{W}^l) \tilde{H}_{(n-1)}^l, (\frac{1_{n-1}^T}{n-1} - \tilde{e}_i^T \tilde{W}^{l'}) \tilde{H}_{(n-1)}^{l'} \rangle$$

$$\leq \sum_{l' \neq l}^{t-1} \mathbb{E} \parallel \frac{1_{n-1}^T}{n-1} - \tilde{e}_i^T \tilde{W}^l \parallel \cdot \parallel \tilde{H}_{(n-1)}^l \parallel \cdot \parallel \frac{1_{n-1}^T}{n-1} - \tilde{e}_i^T \tilde{W}^{l'} \parallel \cdot \parallel \tilde{H}_{(n-1)}^{l'} \parallel$$

$$\leq \rho_2^2 \sum_{l' \neq l}^{t-1} \mathbb{E} (\frac{\parallel \tilde{H}_{(n-1)}^l \parallel^2 + \parallel \tilde{H}_{(n-1)}^{l'} \parallel^2}{2})$$

$$= \rho_2^2 \sum_{l' \neq l}^{t-1} \mathbb{E} \parallel \tilde{H}_{(n-1)}^l \parallel^2 \quad (22)$$

Substitute Equation (21) and (22) into Equation (20) to get

$$\mathbb{E} \parallel \tilde{\theta}_i^t - \frac{1}{n-1} \sum_{i=1}^{n-1} \tilde{\theta}_i^t \parallel^2$$

$$\leq 2\eta^2 \rho_2^2 \sigma_1^2 (n-1)t + 12\eta^2 \rho_2^2 \sigma_2^2 (n-1)t +$$

$$12\eta^2 L^2 \rho_2^2 \sum_{l=0}^{t-1} \sum_{i=1}^{n-1} \mathbb{E} \parallel \tilde{\theta}_i^l - \frac{1}{n-1} \sum_{i=1}^{n-1} \tilde{\theta}_i^l \parallel^2 + 12\eta^2 \rho_2^2 (n-1) \sum_{l=0}^{t-1} \mathbb{E} \parallel \nabla f(\frac{1}{n-1} \sum_{i=1}^{n-1} \tilde{\theta}_i^l) \parallel^2 \quad (23)$$

Summing over $t = 0$ to $t_1 - 1$ and $i = 1$ to $n - 1$ yields:

$$\sum_{t=0}^{t_1-1} \sum_{i=1}^{n-1} \mathbb{E} \parallel \tilde{\theta}_i^t - \frac{1}{n-1} \sum_{i=1}^{n-1} \tilde{\theta}_i^t \parallel^2$$

$$\leq 2\eta^2 \rho_2^2 \sigma_1^2 (n-1)^2 t_1^2 + 12\eta^2 \rho_2^2 \sigma_2^2 (n-1)^2 t_1^2 +$$

$$12\eta^2 L^2 \rho_2^2 (n-1)t_1 \sum_{t=0}^{t_1-1} \sum_{i=1}^{n-1} \mathbb{E} \parallel \tilde{\theta}_i^t - \frac{1}{n-1} \sum_{i=1}^{n-1} \tilde{\theta}_i^t \parallel^2 + 12\eta^2 \rho_2^2 (n-1)^2 t_1 \sum_{t=0}^{t_1-1} \mathbb{E} \parallel \nabla f(\frac{1}{n-1} \sum_{i=1}^{n-1} \tilde{\theta}_i^t) \parallel^2 \quad (24)$$

Therefore, it holds that

$$(1 - 12\eta^2 L^2 \rho_2^2 (n-1)t_1) \sum_{t=0}^{t_1-1} \sum_{i=1}^{n-1} \mathbb{E} \parallel \tilde{\theta}_i^t - \frac{1}{n-1} \sum_{i=1}^{n-1} \tilde{\theta}_i^t \parallel^2$$

$$\leq 2\eta^2 \rho_2^2 \sigma_1^2 (n-1)^2 t_1^2 + 12\eta^2 \rho_2^2 \sigma_2^2 (n-1)^2 t_1^2 + 12\eta^2 \rho_2^2 (n-1)^2 t_1 \sum_{t=0}^{t_1-1} \mathbb{E} \parallel \nabla f(\frac{1}{n-1} \sum_{i=1}^{n-1} \tilde{\theta}_i^t) \parallel^2 \tag{25}$$

Bound $\mathbb{E} \parallel \theta_i^t - \frac{1}{n} \sum_{i=1}^{n} \theta_i^t \parallel^2$:

$$\mathbb{E} \parallel \theta_i^t - \frac{1}{n} \sum_{i=1}^{n} \theta_i^t \parallel^2$$

$$\leq \mathbb{E} \parallel \eta \sum_{l=0}^{t-1} (\frac{1}{n} \sum_{j=1}^{n} \nabla F_j(\theta_j^l, \xi_j^l) - \sum_{j=1}^{n} W_{ij}^l \nabla F_j(\theta_j^l, \xi_j^l)) \parallel^2$$

$$= \eta^2 \mathbb{E} \parallel \sum_{l=0}^{t-1} (\frac{1_n^T}{n} - e_i^T W^l)(G_{(n)}^l - H_{(n)}^l + H_{(n)}^l) \parallel^2$$

$$\leq 2\eta^2 \sum_{l=0}^{t-1} \mathbb{E} \parallel (\frac{1_n^T}{n} - e_i^T W^l)(G_{(n)}^l - H_{(n)}^l) \parallel^2 + 2\eta^2 \sum_{l=0}^{t-1} \mathbb{E} \parallel (\frac{1_n^T}{n} - e_i^T W^l)H_{(n)}^l \parallel^2 +$$

$$2\eta^2 \sum_{l' \neq l}^{t-1} \mathbb{E} \langle (\frac{1_n^T}{n} - e_i^T W^l)H_{(n)}^l, (\frac{1_n^T}{n} - e_i^T W^{l'})H_{(n)}^{l'} \rangle$$

$$\leq 2\eta^2 \rho_1^2 \sigma_1^2 nt + 4\eta^2 \rho_1^2 \sum_{l=0}^{t-1} \mathbb{E} \parallel H_{(n)}^l \parallel^2 \tag{26}$$

And

$$4\eta^2 \rho_1^2 \sum_{l=0}^{t-1} \mathbb{E} \parallel H_{(n)}^l \parallel^2$$

$$\leq 4\eta^2 \rho_1^2 \sum_{l=0}^{t-1} \mathbb{E} \parallel \nabla f_i(\theta_i^l) - \nabla f_i(\frac{1}{n} \sum_{i=1}^{n} \theta_i^l) + f_i(\frac{1}{n} \sum_{i=1}^{n} \theta_i^l) - \nabla f(\frac{1}{n} \sum_{i=1}^{n} \theta_i^l) + \nabla f(\frac{1}{n} \sum_{i=1}^{n} \theta_i^l) \parallel^2$$

$$\leq 12\eta^2 \rho_1^2 L^2 \sum_{l=0}^{t-1} \sum_{i=1}^{n} \mathbb{E} \parallel \theta_i^l - \frac{1}{n} \sum_{i=1}^{n} \theta_i^l \parallel^2 + 12\eta^2 \rho_1^2 \sigma_2^2 nt + 12\eta^2 \rho_1^2 n \sum_{l=0}^{t-1} \mathbb{E} \parallel \nabla f(\frac{1}{n} \sum_{i=1}^{n} \theta_i^l) \parallel^2 \tag{27}$$

Substitute Equation (27) into Equation (26) to get

$$\mathbb{E} \parallel \theta_i^t - \frac{1}{n} \sum_{i=1}^{n} \theta_i^t \parallel^2$$

$$\leq 2\eta^2 \rho_1^2 \sigma_1^2 nt + 12\eta^2 \rho_1^2 \sigma_2^2 nt + 12\eta^2 \rho_1^2 L^2 \sum_{l=0}^{t-1} \sum_{i=1}^{n} \mathbb{E} \parallel \theta_i^l - \frac{1}{n} \sum_{i=1}^{n} \theta_i^l \parallel^2 + 12\eta^2 \rho_1^2 n \sum_{l=0}^{t-1} \mathbb{E} \parallel \nabla f(\frac{1}{n} \sum_{i=1}^{n} \theta_i^l) \parallel^2 \tag{28}$$

Summing over $t = 0$ to $t_1 - 1$ and $i = 1$ to $n$ yields:

$$\sum_{t=0}^{t_1-1} \sum_{i=1}^{n} \mathbb{E} \parallel \theta_i^t - \frac{1}{n} \sum_{i=1}^{n} \theta_i^t \parallel^2$$

$$\leq 2\eta^2 \rho_1^2 \sigma_1^2 n^2 t_1^2 + 12\eta^2 \rho_1^2 \sigma_2^2 n^2 t_1^2 + 12\eta^2 \rho_1^2 L^2 nt_1 \sum_{t=0}^{t_1-1} \sum_{i=1}^{n} \mathbb{E} \parallel \theta_i^t - \frac{1}{n} \sum_{i=1}^{n} \theta_i^t \parallel^2 + 12\eta^2 \rho_1^2 n^2 t_1 \sum_{t=0}^{t_1-1} \mathbb{E} \parallel \nabla f(\frac{1}{n} \sum_{i=1}^{n} \theta_i^t) \parallel^2 \tag{29}$$

Therefore, it holds that

$$(1 - 12\eta^2 L^2 \rho_1^2 nt_1) \sum_{t=0}^{t_1-1} \sum_{i=1}^{n} \mathbb{E} \parallel \theta_i^t - \frac{1}{n} \sum_{i=1}^{n} \theta_i^t \parallel^2$$

$$\leq 2\eta^2 \rho_1^2 \sigma_1^2 n^2 t_1^2 + 12\eta^2 \rho_1^2 \sigma_2^2 n^2 t_1^2 + 12\eta^2 \rho_1^2 n^2 t_1 \sum_{t=0}^{t_1-1} \mathbb{E} \parallel \nabla f(\frac{1}{n}\sum_{i=1}^{n} \theta_i^t) \parallel^2 \qquad (30)$$

According to Equation (18), we have

$$\sum_{t=0}^{t_1-1} \mathbb{E} \parallel \frac{1}{n-1}\sum_{i=1}^{n-1} \theta_i^t - \frac{1}{n-1}\sum_{i=1}^{n-1} \tilde{\theta}_i^t \parallel^2 + \mathbb{E} \parallel \frac{1}{n-1}\sum_{i=1}^{n-1} \theta_i^{t_1} - \frac{1}{n-1}\sum_{i=1}^{n-1} \tilde{\theta}_i^{t_1} \parallel^2$$

$$\leq 24\eta^2 t_1^3 \sigma_1^2 + \frac{12\eta^2 L^2 t_1^2}{n-1} \sum_{t=0}^{t_1-1}\sum_{i=1}^{n-1} \mathbb{E} \parallel \tilde{\theta}_i^t - \frac{1}{n-1}\sum_{i=1}^{n-1} \tilde{\theta}_i^t \parallel^2 + 12\eta^2 L^2 t_1^2 \sum_{t=0}^{t_1-1} \mathbb{E} \parallel \frac{1}{n-1}\sum_{i=1}^{n-1} \tilde{\theta}_i^t - \frac{1}{n-1}\sum_{i=1}^{n-1} \theta_i^t \parallel^2 +$$

$$\frac{12\eta^2 L^2 t_1^2}{n-1} \sum_{t=0}^{t_1-1}\sum_{i=1}^{n} \mathbb{E} \parallel \theta_i^t - \frac{1}{n}\sum_{i=1}^{n} \theta_i^t \parallel^2 + \frac{4\eta^2(1+\rho_1^2)t_1^2}{(n-1)^2} \sum_{t=0}^{t_1-1} \mathbb{E} \parallel G_{(n)}^t - H_{(n)}^t + H_{(n)}^t \parallel^2 \qquad (31)$$

If

$$1 - 12\eta^2 L^2 t_1^2 \geq 0 \Rightarrow \eta \leq \sqrt{\frac{1}{12 L^2 t_1^2}}$$

we have

$$\mathbb{E} \parallel \frac{1}{n-1}\sum_{i=1}^{n-1} \theta_i^{t_1} - \frac{1}{n-1}\sum_{i=1}^{n-1} \tilde{\theta}_i^{t_1} \parallel^2$$

$$\leq 24\eta^2 t_1^3 \sigma_1^2 + \frac{12\eta^2 L^2 t_1^2}{n-1} \sum_{t=0}^{t_1-1}\sum_{i=1}^{n-1} \mathbb{E} \parallel \tilde{\theta}_i^t - \frac{1}{n-1}\sum_{i=1}^{n-1} \tilde{\theta}_i^t \parallel^2 + \frac{12\eta^2 L^2 t_1^2}{n-1} \sum_{t=0}^{t_1-1}\sum_{i=1}^{n} \mathbb{E} \parallel \theta_i^t - \frac{1}{n}\sum_{i=1}^{n} \theta_i^t \parallel^2 +$$

$$\frac{4\eta^2(1+\rho_1^2)t_1^2}{(n-1)^2} \sum_{t=0}^{t_1-1} \mathbb{E} \parallel G_{(n)}^t - H_{(n)}^t + H_{(n)}^t \parallel^2 \qquad (32)$$

Then we need to explore the relationship between $\parallel \theta_i^t - \frac{1}{n}\sum_{i=1}^{n} \theta_i^t \parallel^2$ and $\parallel \nabla f(\frac{1}{n}\sum_{i=1}^{n} \theta_i^t) \parallel^2$:

$$\theta_i^t = \theta_i^0 - \eta \sum_{l=0}^{t-1}\sum_{j=1}^{n} W_{ij}^l \nabla F_j(\theta_j^l, \xi_j^l)$$

$$\frac{1}{n}\sum_{i=1}^{n} \theta_i^t = \theta_i^0 - \frac{\eta}{n} \sum_{l=0}^{t-1}\sum_{j=1}^{n} \nabla F_j(\theta_j^l, \xi_j^l)$$

$$\frac{1}{n}\sum_{i=1}^{n} \theta_i^{t-1} = \theta_i^0 - \frac{\eta}{n} \sum_{l=0}^{t-2}\sum_{j=1}^{n} \nabla F_j(\theta_j^l, \xi_j^l)$$

Based on Assumption 4.5, we can derive the following:

$$\mathbb{E} f(\frac{1}{n}\sum_{i=1}^{n} \theta_i^t) - \mathbb{E} f(\frac{1}{n}\sum_{i=1}^{n} \theta_i^{t-1})$$

$$\leq \underbrace{\mathbb{E}\langle -\frac{\eta}{n}\sum_{i=1}^{n} \nabla F_i(\theta_i^{t-1}, \xi_i^{t-1}), \nabla f(\frac{1}{n}\sum_{i=1}^{n} \theta_i^{t-1})\rangle}_{B_1} + \underbrace{\frac{L}{2}\mathbb{E} \parallel \frac{\eta}{n}\sum_{i=1}^{n} \nabla F_i(\theta_i^{t-1}, \xi_i^{t-1}) \parallel^2}_{B_2}$$

$$B_1 = \mathbb{E}\langle -\frac{\eta}{n}\sum_{i=1}^{n} \nabla f_i(\theta_i^{t-1}), \nabla f(\frac{1}{n}\sum_{i=1}^{n} \theta_i^{t-1})\rangle$$

$$= -\frac{\eta}{2}(\mathbb{E} \| \frac{1}{n} \sum_{i=1}^{n} \nabla f_i(\theta_i^{t-1}) \|^2 + \mathbb{E} \| \nabla f(\frac{1}{n} \sum_{i=1}^{n} \theta_i^{t-1}) \|^2) + \frac{\eta}{2} \mathbb{E} \| \frac{1}{n} \sum_{i=1}^{n} \nabla f_i(\theta_i^{t-1}) - \nabla f(\frac{1}{n} \sum_{i=1}^{n} \theta_i^{t-1}) \|^2$$

$$B_2 = \frac{L}{2} \mathbb{E} \| \frac{\eta}{n} \sum_{i=1}^{n} (\nabla F_i(\theta_i^{t-1}, \xi_i^{t-1}) - \nabla f_i(\theta_i^{t-1}) + \nabla f_i(\theta_i^{t-1})) \|^2$$

$$\leq \frac{\eta^2 L \sigma_1^2}{2n} + \frac{\eta^2 L}{2} \mathbb{E} \| \frac{1}{n} \sum_{i=1}^{n} \nabla f_i(\theta_i^{t-1}) \|^2$$

Therefore, it holds that

$$\mathbb{E}f(\frac{1}{n} \sum_{i=1}^{n} \theta_i^t) - \mathbb{E}f(\frac{1}{n} \sum_{i=1}^{n} \theta_i^{t-1})$$

$$\leq \frac{\eta^2 L \sigma_1^2}{2n} + \frac{\eta L^2}{2n} \sum_{i=1}^{n} \mathbb{E} \| \theta_i^{t-1} - \frac{1}{n} \sum_{i=1}^{n} \theta_i^{t-1} \|^2 - \frac{\eta(1 - \eta L)}{2} \mathbb{E} \| \frac{1}{n} \sum_{i=1}^{n} \nabla f_i(\theta_i^{t-1}) \|^2 - \frac{\eta}{2} \mathbb{E} \| \nabla f(\frac{1}{n} \sum_{i=1}^{n} \theta_i^{t-1}) \|^2 \quad (33)$$

Summing over $t = 1$ to $t_1$ yields:

$$\mathbb{E}f(\frac{1}{n} \sum_{i=1}^{n} \theta_i^{t_1}) - \mathbb{E}f(\frac{1}{n} \sum_{i=1}^{n} \theta_i^0)$$

$$\leq \frac{\eta^2 L \sigma_1^2 t_1}{2n} + \frac{\eta L^2}{2n} \sum_{t=0}^{t_1-1} \sum_{i=1}^{n} \mathbb{E} \| \theta_i^t - \frac{1}{n} \sum_{i=1}^{n} \theta_i^t \|^2 - \frac{\eta(1 - \eta L)}{2} \sum_{t=0}^{t_1-1} \mathbb{E} \| \frac{1}{n} \sum_{i=1}^{n} \nabla f_i(\theta_i^t) \|^2 - \frac{\eta}{2} \sum_{t=0}^{t_1-1} \mathbb{E} \| \nabla f(\frac{1}{n} \sum_{i=1}^{n} \theta_i^t) \|^2$$

$$(34)$$

Then we have

$$\frac{\eta}{2} \sum_{t=0}^{t_1-1} \mathbb{E} \| \nabla f(\frac{1}{n} \sum_{i=1}^{n} \theta_i^t) \|^2 + \frac{\eta(1 - \eta L)}{2} \sum_{t=0}^{t_1-1} \mathbb{E} \| \frac{1}{n} \sum_{i=1}^{n} \nabla f_i(\theta_i^t) \|^2$$

$$\leq \mathbb{E}f(\frac{1}{n} \sum_{i=1}^{n} \theta_i^0) - \mathbb{E}f(\frac{1}{n} \sum_{i=1}^{n} \theta_i^{t_1}) + \frac{\eta L^2}{2n} \sum_{t=0}^{t_1-1} \sum_{i=1}^{n} \mathbb{E} \| \theta_i^t - \frac{1}{n} \sum_{i=1}^{n} \theta_i^t \|^2 + \frac{\eta^2 L \sigma_1^2 t_1}{2n} \quad (35)$$

If the above condition about step size $\eta$ is satisfied, that is, $\eta \leq \sqrt{\frac{1}{12 L^2 t_1^2}}$, then it holds that

$$\sum_{t=0}^{t_1-1} \mathbb{E} \| \nabla f(\frac{1}{n} \sum_{i=1}^{n} \theta_i^t) \|^2 \leq \frac{\eta}{2}\big(\mathbb{E}f(\frac{1}{n} \sum_{i=1}^{n} \theta_i^0) - \mathbb{E}f(\frac{1}{n} \sum_{i=1}^{n} \theta_i^{t_1})\big) + \frac{L^2}{n} \sum_{t=0}^{t_1-1} \sum_{i=1}^{n} \mathbb{E} \| \theta_i^t - \frac{1}{n} \sum_{i=1}^{n} \theta_i^t \|^2 + \frac{\eta L \sigma_1^2 t_1}{n} \quad (36)$$

Then we can further derive the Equation (30) as follows:

$$(1 - 12\eta^2 L^2 \rho_1^2 n t_1) \sum_{t=0}^{t_1-1} \sum_{i=1}^{n} \mathbb{E} \| \theta_i^t - \frac{1}{n} \sum_{i=1}^{n} \theta_i^t \|^2$$

$$\leq 2\eta^2 \rho_1^2 (\sigma_1^2 + 6\sigma_2^2) n^2 t_1^2 + 12\eta^2 \rho_1^2 n^2 t_1 \Big(\frac{\eta}{2}\big(\mathbb{E}f(\frac{1}{n} \sum_{i=1}^{n} \theta_i^0) - \mathbb{E}f(\frac{1}{n} \sum_{i=1}^{n} \theta_i^{t_1})\big) + \frac{L^2}{n} \sum_{t=0}^{t_1-1} \sum_{i=1}^{n} \mathbb{E} \| \theta_i^t - \frac{1}{n} \sum_{i=1}^{n} \theta_i^t \|^2 + \frac{\eta L \sigma_1^2 t_1}{n}\Big)$$

$$(37)$$

Therefore, it holds that

$$(1 - 24\eta^2 L^2 \rho_1^2 n t_1) \sum_{t=0}^{t_1-1} \sum_{i=1}^{n} \mathbb{E} \| \theta_i^t - \frac{1}{n} \sum_{i=1}^{n} \theta_i^t \|^2$$

$$\leq 2\eta^2 \rho_1^2 \sigma_1^2 n^2 t_1^2 + 12\eta^2 \rho_1^2 \sigma_2^2 n^2 t_1^2 + 12\eta^3 L \rho_1^2 \sigma_1^2 n t_1^2 + 24\eta \rho_1^2 n^2 t_1 \big(\mathbb{E}f(\frac{1}{n} \sum_{i=1}^{n} \theta_i^0) - \mathbb{E}f(\frac{1}{n} \sum_{i=1}^{n} \theta_i^{t_1})\big) \quad (38)$$

Next, we need to explore the relationship between $\| \tilde{\theta}_i^t - \frac{1}{n-1} \sum_{i=1}^{n-1} \tilde{\theta}_i^t \|^2$ and $\| \nabla f(\frac{1}{n-1} \sum_{i=1}^{n-1} \tilde{\theta}_i^t) \|^2$:

$$\tilde{\theta}_i^t = \tilde{\theta}_i^0 - \eta \sum_{l=0}^{t-1} \sum_{j=1}^{n} \tilde{W}_{ij}^l \nabla F_j(\tilde{\theta}_j^l, \tilde{\xi}_j^l)$$

$$\frac{1}{n-1} \sum_{i=1}^{n-1} \tilde{\theta}_i^t = \tilde{\theta}_i^0 - \frac{\eta}{n-1} \sum_{l=0}^{t-1} \sum_{j=1}^{n-1} \nabla F_j(\tilde{\theta}_j^l, \tilde{\xi}_j^l)$$

$$\frac{1}{n-1} \sum_{i=1}^{n-1} \tilde{\theta}_i^{t-1} = \tilde{\theta}_i^0 - \frac{\eta}{n-1} \sum_{l=0}^{t-2} \sum_{j=1}^{n-1} \nabla F_j(\tilde{\theta}_j^l, \tilde{\xi}_j^l)$$

Based on Assumption 4.5, we can derive the following:

$$\mathbb{E} f(\frac{1}{n-1} \sum_{i=1}^{n-1} \tilde{\theta}_i^t) - \mathbb{E} f(\frac{1}{n-1} \sum_{i=1}^{n-1} \tilde{\theta}_i^{t-1})$$

$$\leq \underbrace{\mathbb{E} \langle -\frac{\eta}{n-1} \sum_{i=1}^{n-1} \nabla F_i(\tilde{\theta}_i^{t-1}, \tilde{\xi}_i^{t-1}), \nabla f(\frac{1}{n-1} \sum_{i=1}^{n-1} \tilde{\theta}_i^{t-1}) \rangle}_{B_3} + \underbrace{\frac{L}{2} \mathbb{E} \| \frac{\eta}{n-1} \sum_{i=1}^{n-1} \nabla F_i(\tilde{\theta}_i^{t-1}, \tilde{\xi}_i^{t-1}) \|^2}_{B_4}$$

$$B_3 = \mathbb{E} \langle -\frac{\eta}{n-1} \sum_{i=1}^{n-1} \nabla f_i(\tilde{\theta}_i^{t-1}), \nabla f(\frac{1}{n-1} \sum_{i=1}^{n-1} \tilde{\theta}_i^{t-1}) \rangle$$

$$= -\frac{\eta}{2} \mathbb{E} \| \frac{1}{n-1} \sum_{i=1}^{n-1} \nabla f_i(\tilde{\theta}_i^{t-1}) \|^2 - \frac{\eta}{2} \mathbb{E} \| \nabla f(\frac{1}{n-1} \sum_{i=1}^{n-1} \tilde{\theta}_i^{t-1}) \|^2$$

$$+ \frac{\eta}{2} \mathbb{E} \| \frac{1}{n-1} \sum_{i=1}^{n-1} \nabla f_i(\tilde{\theta}_i^{t-1}) - \nabla f(\frac{1}{n-1} \sum_{i=1}^{n-1} \tilde{\theta}_i^{t-1}) \|^2$$

$$\leq -\frac{\eta}{2} \mathbb{E} \| \frac{1}{n-1} \sum_{i=1}^{n-1} \nabla f_i(\tilde{\theta}_i^{t-1}) \|^2 - \frac{\eta}{2} \mathbb{E} \| \nabla f(\frac{1}{n-1} \sum_{i=1}^{n-1} \tilde{\theta}_i^{t-1}) \|^2$$

$$+ \frac{\eta}{2} \mathbb{E} \| \frac{1}{n-1} \sum_{i=1}^{n-1} \left( \nabla f_i(\tilde{\theta}_i^{t-1}) - \nabla f_i(\frac{1}{n-1} \sum_{i=1}^{n-1} \tilde{\theta}_i^{t-1}) + \nabla f_i(\frac{1}{n-1} \sum_{i=1}^{n-1} \tilde{\theta}_i^{t-1}) - \nabla f(\frac{1}{n-1} \sum_{i=1}^{n-1} \tilde{\theta}_i^{t-1}) \right) \|^2$$

$$\leq -\frac{\eta}{2} \mathbb{E} \| \frac{1}{n-1} \sum_{i=1}^{n-1} \nabla f_i(\tilde{\theta}_i^{t-1}) \|^2 - \frac{\eta}{2} \mathbb{E} \| \nabla f(\frac{1}{n-1} \sum_{i=1}^{n-1} \tilde{\theta}_i^{t-1}) \|^2 + \eta \sigma_2^2 + \frac{\eta L^2}{n-1} \sum_{i=1}^{n-1} \mathbb{E} \| \tilde{\theta}_i^{t-1} - \frac{1}{n-1} \sum_{i=1}^{n-1} \tilde{\theta}_i^{t-1} \|^2$$

$$B_4 = \frac{L}{2} \mathbb{E} \| \frac{\eta}{n-1} \sum_{i=1}^{n-1} (\nabla F_i(\tilde{\theta}_i^{t-1}, \tilde{\xi}_i^{t-1}) - \nabla f_i(\tilde{\theta}_i^{t-1}) + \nabla f_i(\tilde{\theta}_i^{t-1})) \|^2$$

$$\leq \frac{\eta^2 L \sigma_1^2}{2(n-1)} + \frac{\eta^2 L}{2} \mathbb{E} \| \frac{1}{n-1} \sum_{i=1}^{n-1} \nabla f_i(\tilde{\theta}_i^{t-1}) \|^2$$

Therefore, it holds that

$$\mathbb{E} f(\frac{1}{n-1} \sum_{i=1}^{n-1} \tilde{\theta}_i^t) - \mathbb{E} f(\frac{1}{n-1} \sum_{i=1}^{n-1} \tilde{\theta}_i^{t-1})$$

$$\leq -\frac{\eta(1-\eta L)}{2} \mathbb{E} \| \frac{1}{n-1} \sum_{i=1}^{n-1} \nabla f_i(\tilde{\theta}_i^{t-1}) \|^2 - \frac{\eta}{2} \mathbb{E} \| \nabla f(\frac{1}{n-1} \sum_{i=1}^{n-1} \tilde{\theta}_i^{t-1}) \|^2 + \frac{\eta^2 L \sigma_1^2}{2(n-1)} + \eta \sigma_2^2$$

$$+ \frac{\eta L^2}{n-1} \sum_{i=1}^{n-1} \mathbb{E} \parallel \tilde{\theta}_i^{t-1} - \frac{1}{n-1} \sum_{i=1}^{n-1} \tilde{\theta}_i^{t-1} \parallel^2 \tag{39}$$

Summing over $t = 1$ to $t_1$ yields:

$$\mathbb{E}f(\frac{1}{n-1} \sum_{i=1}^{n-1} \tilde{\theta}_i^{t_1}) - \mathbb{E}f(\frac{1}{n-1} \sum_{i=1}^{n-1} \tilde{\theta}_i^{0})$$

$$\leq - \frac{\eta(1-\eta L)}{2} \sum_{t=0}^{t_1-1} \mathbb{E} \parallel \frac{1}{n-1} \sum_{i=1}^{n-1} \nabla f_i(\tilde{\theta}_i^t) \parallel^2 - \frac{\eta}{2} \sum_{t=0}^{t_1-1} \mathbb{E} \parallel \nabla f(\frac{1}{n-1} \sum_{i=1}^{n-1} \tilde{\theta}_i^t) \parallel^2 + \frac{\eta^2 L \sigma_1^2 t_1}{2(n-1)} + \eta \sigma_2^2 t_1$$

$$+ \frac{\eta L^2}{n-1} \sum_{t=0}^{t_1-1} \sum_{i=1}^{n-1} \mathbb{E} \parallel \tilde{\theta}_i^t - \frac{1}{n-1} \sum_{i=1}^{n-1} \tilde{\theta}_i^t \parallel^2 \tag{40}$$

Then we have

$$\frac{\eta}{2} \sum_{t=0}^{t_1-1} \mathbb{E} \parallel \nabla f(\frac{1}{n-1} \sum_{i=1}^{n-1} \tilde{\theta}_i^t) \parallel^2 \frac{\eta(1-\eta L)}{2} \sum_{t=0}^{t_1-1} \mathbb{E} \parallel \frac{1}{n-1} \sum_{i=1}^{n-1} \nabla f_i(\tilde{\theta}_i^t) \parallel^2$$

$$\leq \mathbb{E}f(\frac{1}{n-1} \sum_{i=1}^{n-1} \tilde{\theta}_i^{0}) - \mathbb{E}f(\frac{1}{n-1} \sum_{i=1}^{n-1} \tilde{\theta}_i^{t_1}) + \frac{\eta^2 L \sigma_1^2 t_1}{2(n-1)} + \eta \sigma_2^2 t_1 + \frac{\eta L^2}{n-1} \sum_{t=0}^{t_1-1} \sum_{i=1}^{n-1} \mathbb{E} \parallel \tilde{\theta}_i^t - \frac{1}{n-1} \sum_{i=1}^{n-1} \tilde{\theta}_i^t \parallel^2 \tag{41}$$

Similarly, if the above condition about step size $\eta$ is satisfied, that is, $\eta \leq \sqrt{\frac{1}{12L^2 t_1^2}}$, then it holds that

$$\sum_{t=0}^{t_1-1} \mathbb{E} \parallel \nabla f(\frac{1}{n-1} \sum_{i=1}^{n-1} \tilde{\theta}_i^t) \parallel^2$$

$$\leq \frac{2}{\eta} \Big( \mathbb{E}f(\frac{1}{n-1} \sum_{i=1}^{n-1} \tilde{\theta}_i^{0}) - \mathbb{E}f(\frac{1}{n-1} \sum_{i=1}^{n-1} \tilde{\theta}_i^{t_1}) \Big) + \frac{\eta L \sigma_1^2 t_1}{n-1} + 2\sigma_2^2 t_1 + \frac{2L^2}{n-1} \sum_{t=0}^{t_1-1} \sum_{i=1}^{n-1} \mathbb{E} \parallel \tilde{\theta}_i^t - \frac{1}{n-1} \sum_{i=1}^{n-1} \tilde{\theta}_i^t \parallel^2 \tag{42}$$

Then we can further derive the Equation (25) as follows:

$$(1 - 12\eta^2 L^2 \rho_2^2 (n-1) t_1) \sum_{t=0}^{t_1-1} \sum_{i=1}^{n-1} \mathbb{E} \parallel \tilde{\theta}_i^t - \frac{1}{n-1} \sum_{i=1}^{n-1} \tilde{\theta}_i^t \parallel^2$$

$$\leq 2\eta^2 \rho_2^2 \sigma_1^2 (n-1)^2 t_1^2 + 12\eta^2 \rho_2^2 \sigma_2^2 (n-1)^2 t_1^2 + 12\eta^2 \rho_2^2 (n-1)^2 t_1 \Big( \frac{2}{\eta} \big( \mathbb{E}f(\frac{1}{n-1} \sum_{i=1}^{n-1} \tilde{\theta}_i^{0}) - \mathbb{E}f(\frac{1}{n-1} \sum_{i=1}^{n-1} \tilde{\theta}_i^{t_1}) \big)$$

$$+ \frac{\eta L \sigma_1^2 t_1}{n-1} + 2\sigma_2^2 t_1 + \frac{2L^2}{n-1} \sum_{t=0}^{t_1-1} \sum_{i=1}^{n-1} \mathbb{E} \parallel \tilde{\theta}_i^t - \frac{1}{n-1} \sum_{i=1}^{n-1} \tilde{\theta}_i^t \parallel^2 \Big) \tag{43}$$

Therefore, it holds that

$$(1 - 36\eta^2 L^2 \rho_2^2 (n-1) t_1) \sum_{t=0}^{t_1-1} \sum_{i=1}^{n-1} \mathbb{E} \parallel \tilde{\theta}_i^t - \frac{1}{n-1} \sum_{i=1}^{n-1} \tilde{\theta}_i^t \parallel^2$$

$$\leq 2\eta^2 \rho_2^2 \sigma_1^2 (n-1)^2 t_1^2 + 12\eta^2 \rho_2^2 \sigma_2^2 (n-1)^2 t_1^2 + 12\eta^3 L \rho_2^2 \sigma_1^2 (n-1) t_1^2 + 24\eta^2 \rho_2^2 \sigma_2^2 (n-1)^2 t_1^2 +$$

$$24\eta \rho_2^2 (n-1)^2 t_1 \big( \mathbb{E}f(\frac{1}{n-1} \sum_{i=1}^{n-1} \tilde{\theta}_i^{0}) - \mathbb{E}f(\frac{1}{n-1} \sum_{i=1}^{n-1} \tilde{\theta}_i^{t_1}) \big) \tag{44}$$

Then based on Equation (19), we can further derive the following:

$$\frac{4\eta^2 (1 + \rho_1^2) t_1^2}{(n-1)^2} \sum_{t=0}^{t_1-1} \mathbb{E} \parallel G_{(n)}^t - H_{(n)}^t + H_{(n)}^t \parallel^2$$

$$\leq 2 \sum_{i=1}^{n} \mathbb{E} \parallel \nabla F_i(\theta_i^t, \xi_i^t) - \nabla f_i(\theta_i^t) \parallel^2 +$$

$$2 \sum_{i=1}^{n} \mathbb{E} \parallel \nabla f_i(\theta_i^t) - \nabla f_i(\frac{1}{n} \sum_{i=1}^{n} \theta_i^t) + \nabla f_i(\frac{1}{n} \sum_{i=1}^{n} \theta_i^t) - \nabla f(\frac{1}{n} \sum_{i=1}^{n} \theta_i^t) + \nabla f(\frac{1}{n} \sum_{i=1}^{n} \theta_i^t) \parallel^2$$

$$\leq \frac{8\eta^2(1+\rho_1^2)\sigma_1^2 n t_1^3}{(n-1)^2} + \frac{24\eta^2(1+\rho_1^2)\sigma_2^2 n t_1^3}{(n-1)^2} + \frac{24\eta^2 L^2(1+\rho_1^2)t_1^2}{(n-1)^2} \sum_{t=0}^{t_1-1} \sum_{i=1}^{n} \mathbb{E} \parallel \theta_i^t - \frac{1}{n} \sum_{i=1}^{n} \theta_i^t \parallel^2$$

$$+ \frac{24\eta^2(1+\rho_1^2)n t_1^2}{(n-1)^2} \sum_{t=0}^{t_1-1} \mathbb{E} \parallel \nabla f(\frac{1}{n} \sum_{i=1}^{n} \theta_i^t) \parallel^2 \tag{45}$$

Due to the fact

$$n \geq 2 \quad \Rightarrow \quad \begin{cases} \dfrac{1}{(n-1)^2} \leq \dfrac{2}{n}, \\[2mm] \dfrac{1}{n-1} \leq \dfrac{2}{n}, \\[2mm] \dfrac{n}{(n-1)^2} \leq \dfrac{2}{n-1} \leq \dfrac{4}{n}, \end{cases}$$

we can further derive the Equation (32) as

$$\mathbb{E} \parallel \frac{1}{n-1} \sum_{i=1}^{n-1} \theta_i^{t_1} - \frac{1}{n-1} \sum_{i=1}^{n-1} \tilde{\theta}_i^{t_1} \parallel^2$$

$$\leq 8\eta^2(5+2\rho_1^2)\sigma_1^2 t_1^3 + 48\eta^2(1+\rho_1^2)\sigma_2^2 t_1^3 + \frac{12\eta^2 L^2 t_1^2}{n-1} \sum_{t=0}^{t_1-1} \sum_{i=1}^{n-1} \mathbb{E} \parallel \tilde{\theta}_i^t - \frac{1}{n-1} \sum_{i=1}^{n-1} \tilde{\theta}_i^t \parallel^2 +$$

$$\frac{24\eta^2 L^2(3+2\rho_1^2)t_1^2}{n} \sum_{t=0}^{t_1-1} \sum_{i=1}^{n} \mathbb{E} \parallel \theta_i^t - \frac{1}{n} \sum_{i=1}^{n} \theta_i^t \parallel^2 + \frac{24\eta^2(1+\rho_1^2)n t_1^2}{(n-1)^2} \sum_{t=0}^{t_1-1} \mathbb{E} \parallel \nabla f(\frac{1}{n} \sum_{i=1}^{n} \theta_i^t) \parallel^2 \tag{46}$$

According to Equation (38) and (47), let

$$D_1 = 1 - 24\eta^2 L^2 \rho_1^2 n t_1 > 0 \Rightarrow \eta < \sqrt{\frac{1}{24 L^2 \rho_1^2 n t_1}}$$

$$D_2 = 1 - 36\eta^2 L^2 \rho_2^2 (n-1) t_1 > 0 \Rightarrow \eta < \sqrt{\frac{1}{36 L^2 \rho_2^2 (n-1) t_1}}$$

$$\frac{1}{n} \sum_{t=0}^{t_1-1} \sum_{i=1}^{n} \mathbb{E} \parallel \theta_i^t - \frac{1}{n} \sum_{i=1}^{n} \theta_i^t \parallel^2$$

$$\leq \frac{2\eta^2 \rho_1^2 \sigma_1^2 n t_1^2}{D_1} + \frac{12\eta^2 \rho_1^2 \sigma_2^2 n t_1^2}{D_1} + \frac{12\eta^3 L \rho_1^2 \sigma_1^2 t_1^2}{D_1} + \frac{24\eta \rho_1^2 n t_1}{D_1} \big( \mathbb{E} f(\frac{1}{n} \sum_{i=1}^{n} \theta_i^0) - \mathbb{E} f(\frac{1}{n} \sum_{i=1}^{n} \theta_i^{t_1}) \big) \tag{47}$$

$$\frac{1}{n-1} \sum_{t=0}^{t_1-1} \sum_{i=1}^{n-1} \mathbb{E} \parallel \tilde{\theta}_i^t - \frac{1}{n-1} \sum_{i=1}^{n-1} \tilde{\theta}_i^t \parallel^2$$

$$\leq \frac{2\eta^2 \rho_2^2 \sigma_1^2 (n-1) t_1^2}{D_2} + \frac{12\eta^2 \rho_2^2 \sigma_2^2 (n-1) t_1^2}{D_2} + \frac{12\eta^3 L \rho_2^2 \sigma_1^2 t_1^2}{D_2} + \frac{24\eta^2 \rho_2^2 \sigma_2^2 (n-1) t_1^2}{D_2} +$$

$$\frac{24\eta \rho_2^2 (n-1) t_1}{D_2} \big( \mathbb{E} f(\frac{1}{n-1} \sum_{i=1}^{n-1} \tilde{\theta}_i^0) - \mathbb{E} f(\frac{1}{n-1} \sum_{i=1}^{n-1} \tilde{\theta}_i^{t_1}) \big) \tag{48}$$

Substitute Equation (36), (47) and (48) into Equation (46) to get

$$
\mathbb{E} \parallel \frac{1}{n-1} \sum_{i=1}^{n-1} \theta_i^{t_1} - \frac{1}{n-1} \sum_{i=1}^{n-1} \tilde{\theta}_i^{t_1} \parallel^2
$$

$$
\leq (24\eta^4 L^2 \sigma_1^2 n t_1^4 + 144\eta^5 L^3 \sigma_1^2 t_1^4) \cdot \left( \frac{(6+4\rho_1^2)\rho_1^2}{D_1} + \frac{\rho_2^2}{D_2} + \frac{8(1+\rho_1^2)\rho_1^2}{nD_1} \right) + 8\eta^2(5+2\rho_1^2)\sigma_1^2 t_1^3 + 48\eta^2(1+\rho_1^2)\sigma_2^2 t_1^3 +
$$

$$
144\eta^4 L^2 \sigma_2^2 t_1^4 \cdot \left( \frac{(6+4\rho_1^2)\rho_1^2 n}{D_1} + \frac{3\rho_2^2(n-1)}{D_2} + \frac{16(1+\rho_1^2)\rho_1^2}{D_1} \right) +
$$

$$
\left( \frac{2304\eta^3 L^2 \rho_1^2 (1+\rho_1^2) t_1^3}{D_1} + \frac{576\eta^3 L^2 \rho_1^2 (3+2\rho_1^2) n t_1^3}{D_1} + \frac{192\eta(1+\rho_1^2) t_1^2}{n} \right) \cdot \left( \mathbb{E}f(\theta^0) - \mathbb{E}f(\frac{1}{n}\sum_{i=1}^{n} \theta_i^{t_1}) \right) +
$$

$$
\frac{288\eta^3 L^2 \rho_2^2 (n-1) t_1^3}{D_2} \cdot \left( \mathbb{E}f(\theta^0) - \mathbb{E}f(\frac{1}{n-1}\sum_{i=1}^{n-1} \tilde{\theta}_i^{t_1}) \right) \tag{49}
$$

where $0 < \eta < \min\{\sqrt{\frac{1}{12L^2 t_1^2}}, \sqrt{\frac{1}{24L^2 \rho_1^2 n t_1}}, \sqrt{\frac{1}{36L^2 \rho_2^2 (n-1) t_1}}\}$. Then, using Jensen's inequality, we can directly obtain Lemma B.1.

## D. Proof of Lemma B.2

$$
\mathbb{E} \parallel \frac{1}{n-1} \sum_{i=1}^{n-1} \sum_{t=0}^{t_1-1} p_i^t \delta_i^t \parallel^2
$$

$$
\leq \frac{t_1}{n-1} \sum_{i=1}^{n-1} \sum_{t=0}^{t_1-1} \parallel p_i^t \parallel^2 \mathbb{E} \parallel \delta_i^t \parallel^2
$$

$$
\leq \frac{t_1}{n-1} \sum_{i=1}^{n-1} \sum_{t=0}^{t_1-1} \mathbb{E} \parallel \delta_i^t \parallel^2
$$

$$
= \frac{\eta^2 t_1}{n-1} \sum_{i=1}^{n-1} \sum_{t=0}^{t_1-1} \mathbb{E} \parallel \sum_{j=1}^{n-1} \tilde{W}_{ij}^t \nabla F_j(\theta_j^t, \xi_j^t) - \frac{1}{n-1} \sum_{j=1}^{n-1} \nabla F_j(\theta_j^t, \xi_j^t) + \frac{1}{n} \sum_{j=1}^{n} \nabla F_j(\theta_j^t, \xi_j^t) - \sum_{j=1}^{n} W_{ij}^t \nabla F_j(\theta_j^t, \xi_j^t)
$$

$$
+ \frac{1}{n-1} (\frac{1}{n} \nabla F_j(\theta_j^t, \xi_j^t) - \nabla F_n(\theta_n^t, \xi_n^t)) \parallel^2
$$

$$
\leq \frac{3\eta^2 t_1}{n-1} \sum_{i=1}^{n-1} \sum_{t=0}^{t_1-1} \left( \mathbb{E} \parallel (\tilde{e}_i^T \tilde{W}^t - \frac{1_{n-1}^T}{n-1}) G_{(n-1)}^t \parallel^2 + \mathbb{E} \parallel (e_i^T W^t - \frac{1_n^T}{n}) G_{(n)}^t \parallel^2 + \frac{1}{(n-1)^2} \mathbb{E} \parallel (\frac{1_n^T}{n} - e_n^T) G_{(n)}^t \parallel^2 \right)
$$

$$
\leq 3\eta^2 (\rho_1^2 + \rho_2^2 + \frac{1}{(n-1)^2}) t_1 \sum_{t=0}^{t_1-1} \mathbb{E} \parallel G_{(n)}^t - H_{(n)}^t + H_{(n)}^t \parallel^2 \tag{50}
$$

Substitute Equation (19) and (47) into Equation (50) to get

$$
\mathbb{E} \parallel \frac{1}{n-1} \sum_{i=1}^{n-1} \sum_{t=0}^{t_1-1} p_i^t \delta_i^t \parallel^2
$$

$$
\leq 6 \left( \eta^2 n + 3\eta^3 L + \frac{12\eta^4 L^2 \rho_1^2 n^2 t_1}{D_1} + \frac{72\eta^5 L^3 \rho_1^2 n t_1}{D_1} \right) D_3 \sigma_1^2 t_1^2 + \left( 18\eta^2 n + \frac{432\eta^4 L^2 \rho_1^2 n^2 t_1}{D_2} \right) D_3 \sigma_2^2 t_1^2 +
$$

$$
\left( 36\eta n t_1 + \frac{864\eta^3 L^2 \rho_1^2 n^2 t_1^2}{D_1} \right) D_3 \left( \mathbb{E}f(\theta^0) - \mathbb{E}f(\frac{1}{n}\sum_{i=1}^{n} \theta_i^{t_1}) \right) \tag{51}
$$

where $D_3 = \rho_1^2 + \rho_2^2 + \frac{1}{(n-1)^2}$ and step size $0 < \eta < \min\{\sqrt{\frac{1}{12L^2 t_1^2}}, \sqrt{\frac{1}{24L^2 \rho_1^2 n t_1}}, \sqrt{\frac{1}{36L^2 \rho_2^2 (n-1) t_1}}\}$. Then, using Jensen's inequality, we can directly obtain Lemma B.2 without early stopping.

For an early stopping setup, it holds that

$$
\mathbb{E} \| \frac{1}{n-1} \sum_{i=1}^{n-1} \sum_{t=0}^{t_{1,i}-1} {p_i'}^t \delta_i^t \|^2
$$

$$
\leq \frac{1}{(n-1)^2} \sum_{i=1}^{n-1} (n-1) \sum_{t=0}^{t_{1,i}-1} t_{1,i} \| {p_i'}^t \|^2 \, \mathbb{E} \| \delta_i^t \|^2
$$

$$
\leq \frac{t_1}{n-1} \sum_{i=1}^{n-1} \sum_{t=0}^{t_1-1} \mathbb{E} \| \delta_i^t \|^2 \tag{52}
$$

This result shows the same bound with Equation (50). As a result, Lemma B.2 still holds for the early stopping setup.

## E. Proof of Theorem 4.9

Subtracting Equation (12) from Equation (13) yields

$$
\frac{1}{n-1} \sum_{i=1}^{n-1} \bar{\bar{\theta}}_i^{t_1} - \frac{1}{n-1} \sum_{i=1}^{n-1} \tilde{\theta}_i^{t_1} = \frac{1}{n-1} \sum_{i=1}^{n-1} \theta_i^{t_1} - \frac{1}{n-1} \sum_{i=1}^{n-1} \tilde{\theta}_i^{t_1} - \frac{1}{n-1} \sum_{i=1}^{n-1} \sum_{t=0}^{t_1-1} p_i^t \delta_i^t \tag{53}
$$

If $0 < \eta < \min\{\sqrt{\frac{1}{12L^2 t_1^2}}, \sqrt{\frac{1}{24L^2\rho_1^2 n t_1}}, \sqrt{\frac{1}{36L^2\rho_2^2(n-1)t_1}}\}$, it holds that

$$
\| \frac{1}{n-1} \sum_{i=1}^{n-1} \bar{\bar{\theta}}_i^{t_1} - \frac{1}{n-1} \sum_{i=1}^{n-1} \tilde{\theta}_i^{t_1} \|
$$

$$
\leq \| \frac{1}{n-1} \sum_{i=1}^{n-1} \theta_i^{t_1} - \frac{1}{n-1} \sum_{i=1}^{n-1} \tilde{\theta}_i^{t_1} \| + \| \frac{1}{n-1} \sum_{i=1}^{n-1} \sum_{t=0}^{t_1-1} p_i^t \delta_i^t \|
$$

$$
\leq \sqrt{ \begin{aligned} &(24\eta^4 L^2 \sigma_1^2 n t_1^4 + 144\eta^5 L^3 \sigma_1^2 t_1^4) \cdot (\frac{(6+4\rho_1^2)\rho_1^2}{D_1} + \frac{\rho_2^2}{D_2} + \frac{8(1+\rho_1^2)\rho_1^2}{nD_1}) + 8\eta^2(5 + 2\rho_1^2)\sigma_1^2 t_1^3 + 48\eta^2(1 + \rho_1^2)\sigma_2^2 t_1^3 + \\ &144\eta^4 L^2 \sigma_2^2 t_1^4 \cdot (\frac{(6+4\rho_1^2)\rho_1^2 n}{D_1} + \frac{3\rho_2^2(n-1)}{D_2} + \frac{16(1+\rho_1^2)\rho_1^2}{D_1}) + (\frac{2304\eta^3 L^2 \rho_1^2(1+\rho_1^2)t_1^3}{D_1} + \frac{576\eta^3 L^2 \rho_1^2(3+2\rho_1^2)nt_1^3}{D_1} + \\ &\frac{192\eta(1+\rho_1^2)t_1^2}{n}) \cdot (\mathbb{E}f(\theta^0) - \mathbb{E}f(\frac{1}{n}\sum_{i=1}^{n}\theta_i^{t_1})) + \frac{288\eta^3 L^2 \rho_2^2(n-1)t_1^3}{D_2} \cdot (\mathbb{E}f(\theta^0) - \mathbb{E}f(\frac{1}{n-1}\sum_{i=1}^{n-1}\tilde{\theta}_i^{t_1})) \end{aligned} }
$$

$$
+ \sqrt{ \begin{aligned} &6(\eta^2 n + 3\eta^3 L + \frac{12\eta^4 L^2 \rho_1^2 n^2 t_1}{D_1} + \frac{72\eta^5 L^3 \rho_1^2 n t_1}{D_1})D_3\sigma_1^2 t_1^2 + (18\eta^2 n + \frac{432\eta^4 L^2 \rho_1^2 n^2 t_1}{D_2})D_3\sigma_2^2 t_1^2 + \\ &(36\eta n t_1 + \frac{864\eta^3 L^2 \rho_1^2 n^2 t_1^2}{D_1})D_3(\mathbb{E}f(\theta^0) - \mathbb{E}f(\frac{1}{n}\sum_{i=1}^{n}\theta_i^{t_1})) \end{aligned} }
$$

which completes the proof.

## F. Proof of Corollary 4.10

Corollary 4.10 guarantees the $(\epsilon, \beta)$-Indistinguishability of Algorithm 2, since it is essentially based on the Gaussian mechanism.

From a global perspective, the average model $\frac{1}{n-1}\sum_{i=1}^{n-1}\theta_i^u$ is obtained by our proposed decentralized unlearning algorithm (Algorithm 2) and $\frac{1}{n-1}\sum_{i=1}^{n-1}\check{\theta}_i^{t_1}$ is produced by the retraining algorithm (Algorithm 1).

According to the generation rule of $\check{\theta}_i^{t_1}$ and $\theta_i^u$:

$$
\begin{cases} \check{\theta}_i^{t_1} = \tilde{\theta}_i^{t_1} + z_i, \text{ where } z_i \sim \mathcal{N}(0, (n-1)\sigma^2 \mathbb{I}_d) \\ \theta_i^u = \bar{\bar{\theta}}_i^{t_1} + z_i, \text{ where } z_i \sim \mathcal{N}(0, (n-1)\sigma^2 \mathbb{I}_d) \end{cases}
$$

We can derive that the average models $\frac{1}{n-1}\sum_{i=1}^{n-1}\breve{\theta}_i^{t_1}$ and $\frac{1}{n-1}\sum_{i=1}^{n-1}\theta_i^u$ satisfy:

$$\begin{cases} \dfrac{1}{n-1}\sum_{i=1}^{n-1}\breve{\theta}_i^{t_1} = \dfrac{1}{n-1}\sum_{i=1}^{n-1}\tilde{\theta}_i^{t_1} + z \\[3mm] \dfrac{1}{n-1}\sum_{i=1}^{n-1}\theta_i^u = \dfrac{1}{n-1}\sum_{i=1}^{n-1}\bar{\bar{\theta}}_i^{t_1} + z \end{cases}$$

where $z \sim \mathcal{N}(0, \sigma^2 \mathbb{I}_d)$.

Therefore, the average models $\frac{1}{n-1}\sum_{i=1}^{n-1}\breve{\theta}_i^{t_1}$ and $\frac{1}{n-1}\sum_{i=1}^{n-1}\theta_i^u$ follow the distributions $\frac{1}{n-1}\sum_{i=1}^{n-1}\breve{\theta}_i^{t_1} \sim \mathcal{N}(\frac{1}{n-1}\sum_{i=1}^{n-1}\tilde{\theta}_i^{t_1}, \sigma^2\mathbb{I}_d)$ and $\frac{1}{n-1}\sum_{i=1}^{n-1}\theta_i^u \sim \mathcal{N}(\frac{1}{n-1}\sum_{i=1}^{n-1}\bar{\bar{\theta}}_i^{t_1}, \sigma^2\mathbb{I}_d)$.

What's more, according to Theorem 4.9, we can denote $d_1$ as the upper bound discussed in Theorem 4.9, which satisfies

$$d_1 \geq \sqrt{\begin{aligned} &(24\eta^4 L^2\sigma_1^2 nt_1^4 + 144\eta^5 L^3\sigma_1^2 t_1^4)\cdot\left(\tfrac{(6+4\rho_1^2)\rho_1^2}{D_1} + \tfrac{\rho_2^2}{D_2} + \tfrac{8(1+\rho_1^2)\rho_1^2}{nD_1}\right) + 8\eta^2(5+2\rho_1^2)\sigma_1^2 t_1^3 + 48\eta^2(1+\rho_1^2)\sigma_2^2 t_1^3 + \\ &144\eta^4 L^2\sigma_2^2 t_1^4\cdot\left(\tfrac{(6+4\rho_1^2)\rho_1^2 n}{D_1} + \tfrac{3\rho_2^2(n-1)}{D_2} + \tfrac{16(1+\rho_1^2)\rho_1^2}{D_1}\right) + \left(\tfrac{2304\eta^3 L^2\rho_1^2(1+\rho_1^2)t_1^3}{D_1} + \tfrac{576\eta^3 L^2\rho_1^2(3+2\rho_1^2)nt_1^3}{D_1} + \right. \\ &\left. \tfrac{192\eta(1+\rho_1^2)t_1^2}{n}\right)\cdot\left(\mathbb{E}f(\theta^0) - \mathbb{E}f(\tfrac{1}{n}\sum_{i=1}^{n}\theta_i^{t_1})\right) + \tfrac{288\eta^3 L^2\rho_2^2(n-1)t_1^3}{D_2}\cdot\left(\mathbb{E}f(\theta^0) - \mathbb{E}f(\tfrac{1}{n-1}\sum_{i=1}^{n-1}\tilde{\theta}_i^{t_1})\right) \end{aligned}}$$
$$+ \sqrt{\begin{aligned} &6\left(\eta^2 n + 3\eta^3 L + \tfrac{12\eta^4 L^2\rho_1^2 n^2 t_1}{D_1} + \tfrac{72\eta^5 L^3\rho_1^2 nt_1}{D_1}\right)D_3\sigma_1^2 t_1^2 + \left(18\eta^2 n + \tfrac{432\eta^4 L^2\rho_1^2 n^2 t_1}{D_2}\right)D_3\sigma_2^2 t_1^2 + \\ &\left(36\eta nt_1 + \tfrac{864\eta^3 L^2\rho_1^2 n^2 t_1^2}{D_1}\right)D_3\left(\mathbb{E}f(\theta^0) - \mathbb{E}f(\tfrac{1}{n}\sum_{i=1}^{n}\theta_i^{t_1})\right) \end{aligned}}$$

Based on Gaussian mechanism (Definition 4.3), the average models $\frac{1}{n-1}\sum_{i=1}^{n-1}\breve{\theta}_i^{t_1}$ and $\frac{1}{n-1}\sum_{i=1}^{n-1}\theta_i^u$ are $(\epsilon, \beta)$-Indistinguishability, and thus Algorithm 2 satisfies $(\epsilon, \beta)$-machine unlearning with

$$\sigma = \frac{1}{\sqrt{2}}\cdot\frac{d_1}{\sqrt{\log(1/\beta)+\epsilon} - \sqrt{\log(1/\beta)}}.$$

That completes the proof.

## G. Proof of Theorem 4.11

Based on Equation (10), we have

$$\hat{\theta}_i^t = \hat{\theta}_i^0 - \hat{\eta}\sum_{l=1}^{t}\sum_{j=1}^{n-1}\tilde{W}_{ij}^{t_1+l-1}\nabla F_j(\hat{\theta}_j^{l-1}, \hat{\xi}_j^{l-1})$$

$$\frac{1}{n-1}\sum_{i=1}^{n-1}\hat{\theta}_i^t = \frac{1}{n-1}\sum_{i=1}^{n-1}\hat{\theta}_i^0 - \frac{\hat{\eta}}{n-1}\sum_{l=1}^{t}\sum_{j=1}^{n-1}\nabla F_j(\hat{\theta}_j^{l-1}, \hat{\xi}_j^{l-1})$$

$$\frac{1}{n-1}\sum_{i=1}^{n-1}\hat{\theta}_i^{t-1} = \frac{1}{n-1}\sum_{i=1}^{n-1}\hat{\theta}_i^0 - \frac{\hat{\eta}}{n-1}\sum_{l=1}^{t-1}\sum_{j=1}^{n-1}\nabla F_j(\hat{\theta}_j^{l-1}, \hat{\xi}_j^{l-1})$$

Therefore, it holds that

$$\hat{\theta}_i^t - \frac{1}{n-1}\sum_{i=1}^{n-1}\hat{\theta}_i^t$$

$$=(\hat{\theta}_i^0 - \frac{1}{n-1}\sum_{i=1}^{n-1}\hat{\theta}_i^0) - \hat{\eta}\sum_{l=1}^{t}\big(\sum_{j=1}^{n-1}\tilde{W}_{ij}^{t_1+l-1}\nabla F_j(\hat{\theta}_j^{l-1},\hat{\xi}_j^{l-1}) - \frac{1}{n-1}\sum_{j=1}^{n-1}\nabla F_j(\hat{\theta}_j^{l-1},\hat{\xi}_j^{l-1}))$$

$$=(\theta_i^u - \frac{1}{n-1}\sum_{i=1}^{n-1}\theta_i^u) - \hat{\eta}\sum_{l=1}^{t}(\tilde{e}_i^T\tilde{W}^{t_1+l-1} - \frac{1_{n-1}^T}{n-1})\hat{G}_{(n-1)}^{l-1} \tag{54}$$

$$\frac{1}{n-1}\sum_{i=1}^{n-1}\hat{\theta}_i^t - \frac{1}{n-1}\sum_{i=1}^{n-1}\hat{\theta}_i^{t-1} = -\frac{\hat{\eta}}{n-1}\sum_{j=1}^{n-1}\nabla F_j(\hat{\theta}_j^{t-1},\hat{\xi}_j^{t-1}) \tag{55}$$

According to Assumption 4.5, we have

$$\mathbb{E}\tilde{f}(\frac{1}{n-1}\sum_{i=1}^{n-1}\hat{\theta}_i^t) - \mathbb{E}\tilde{f}(\frac{1}{n-1}\sum_{i=1}^{n-1}\hat{\theta}_i^{t-1})$$

$$\leq \mathbb{E}\langle -\frac{\hat{\eta}}{n-1}\sum_{i=1}^{n-1}\nabla f_i(\hat{\theta}_i^{t-1}), \nabla \tilde{f}(\frac{1}{n-1}\sum_{i=1}^{n-1}\hat{\theta}_i^{t-1})\rangle + \frac{L}{2}\mathbb{E}\parallel -\frac{\hat{\eta}}{n-1}\sum_{i=1}^{n-1}\nabla F_i(\hat{\theta}_i^{t-1},\hat{\xi}_i^{t-1})\parallel^2$$

$$= -\frac{\hat{\eta}}{2}\mathbb{E}\parallel\nabla\tilde{f}(\frac{1}{n-1}\sum_{i=1}^{n-1}\hat{\theta}_i^{t-1})\parallel^2 - \frac{\hat{\eta}}{2}\mathbb{E}\parallel\frac{1}{n-1}\sum_{i=1}^{n-1}\nabla f_i(\hat{\theta}_i^{t-1})\parallel^2$$

$$+ \frac{\hat{\eta}}{2}\mathbb{E}\parallel\nabla\tilde{f}(\frac{1}{n-1}\sum_{i=1}^{n-1}\hat{\theta}_i^{t-1}) - \frac{1}{n-1}\sum_{i=1}^{n-1}\nabla f_i(\hat{\theta}_i^{t-1})\parallel^2$$

$$+ \frac{\hat{\eta}^2 L}{2}\mathbb{E}\parallel\frac{1}{n-1}\sum_{i=1}^{n-1}\nabla F_i(\hat{\theta}_i^{t-1},\hat{\xi}_i^{t-1}) - \frac{1}{n-1}\sum_{i=1}^{n-1}\nabla f_i(\hat{\theta}_i^{t-1}) + \frac{1}{n-1}\sum_{i=1}^{n-1}\nabla f_i(\hat{\theta}_i^{t-1})\parallel^2$$

$$= -\frac{\hat{\eta}}{2}\mathbb{E}\parallel\nabla\tilde{f}(\frac{1}{n-1}\sum_{i=1}^{n-1}\hat{\theta}_i^{t-1})\parallel^2 - \frac{\hat{\eta}}{2}\mathbb{E}\parallel\frac{1}{n-1}\sum_{i=1}^{n-1}\nabla f_i(\hat{\theta}_i^{t-1})\parallel^2$$

$$+ \frac{\hat{\eta}}{2}\mathbb{E}\parallel\frac{1}{n-1}\sum_{i=1}^{n-1}(\nabla f_i(\frac{1}{n-1}\sum_{i=1}^{n-1}\hat{\theta}_i^{t-1}) - \nabla f_i(\hat{\theta}_i^{t-1}))\parallel^2$$

$$+ \frac{\hat{\eta}^2 L}{2}\mathbb{E}\parallel\frac{1}{n-1}\sum_{i=1}^{n-1}(\nabla F_i(\hat{\theta}_i^{t-1},\hat{\xi}_i^{t-1}) - \nabla f_i(\hat{\theta}_i^{t-1}))\parallel^2 + \frac{\hat{\eta}^2 L}{2}\mathbb{E}\parallel\frac{1}{n-1}\sum_{i=1}^{n-1}\nabla f_i(\hat{\theta}_i^{t-1})\parallel^2$$

$$\leq \frac{\hat{\eta}^2 L\sigma_1^2}{2(n-1)} - \frac{\hat{\eta}}{2}\mathbb{E}\parallel\nabla\tilde{f}(\frac{1}{n-1}\sum_{i=1}^{n-1}\hat{\theta}_i^{t-1})\parallel^2 - \frac{\hat{\eta}(1-\hat{\eta}L)}{2}\mathbb{E}\parallel\frac{1}{n-1}\sum_{i=1}^{n-1}\nabla f_i(\hat{\theta}_i^{t-1})\parallel^2$$

$$+ \frac{\hat{\eta}L^2}{2(n-1)}\underbrace{\sum_{i=1}^{n-1}\mathbb{E}\parallel\hat{\theta}_i^{t-1} - \frac{1}{n-1}\sum_{i=1}^{n-1}\hat{\theta}_i^{t-1}\parallel^2}_{B_5} \tag{56}$$

Bound $\sum_{i=1}^{n-1}\mathbb{E}\parallel\hat{\theta}_i^t - \frac{1}{n-1}\sum_{i=1}^{n-1}\hat{\theta}_i^t\parallel^2$:

$$\sum_{i=1}^{n-1}\mathbb{E}\parallel\hat{\theta}_i^t - \frac{1}{n-1}\sum_{i=1}^{n-1}\hat{\theta}_i^t\parallel^2$$

$$= \sum_{i=1}^{n-1}\mathbb{E}\parallel(\theta_i^u - \frac{1}{n-1}\sum_{i=1}^{n-1}\theta_i^u) - \hat{\eta}\sum_{l=1}^{t}(\tilde{e}_i^T\tilde{W}^{t_1+l-1} - \frac{1_{n-1}^T}{n-1})\hat{G}_{(n-1)}^{l-1}\parallel^2$$

$$\leq 2\sum_{i=1}^{n-1}\mathbb{E}\parallel\theta_i^u - \frac{1}{n-1}\sum_{i=1}^{n-1}\theta_i^u\parallel^2 + 4\hat{\eta}^2\sum_{i=1}^{n-1}\mathbb{E}\parallel\sum_{l=1}^{t}(\tilde{e}_i^T\tilde{W}^{t_1+l-1} - \frac{1_{n-1}^T}{n-1})(\hat{G}_{(n-1)}^{l-1} - \hat{H}_{(n-1)}^{l-1})\parallel^2$$

$$+ 4\hat{\eta}^2 \sum_{i=1}^{n-1} \mathbb{E} \parallel \sum_{l=1}^{t} (\tilde{e}_i^T \tilde{W}^{t_1+l-1} - \frac{1_{n-1}^T}{n-1}) \hat{H}_{(n-1)}^{l-1} \parallel^2$$

$$\leq 2 \sum_{i=1}^{n-1} \mathbb{E} \parallel \theta_i^u - \frac{1}{n-1} \sum_{i=1}^{n-1} \theta_i^u \parallel^2 + 4\hat{\eta}^2 \rho_2^2 \sigma_1^2 (n-1)^2 t + 4\hat{\eta}^2 \sum_{i=1}^{n-1} \sum_{l=1}^{t} \mathbb{E} \parallel (\tilde{e}_i^T \tilde{W}^{t_1+l-1} - \frac{1_{n-1}^T}{n-1}) \hat{H}_{(n-1)}^{l-1} \parallel^2 +$$

$$\sum_{l \neq l^*}^{t} \mathbb{E} \langle (\tilde{e}_i^T \tilde{W}^{t_1+l-1} - \frac{1_{n-1}^T}{n-1}) \hat{H}_{(n-1)}^{l-1}, (\tilde{e}_i^T \tilde{W}^{t_1+l^*-1} - \frac{1_{n-1}^T}{n-1}) \hat{H}_{(n-1)}^{l^*-1} \rangle$$

$$\leq 2 \sum_{i=1}^{n-1} \mathbb{E} \parallel \theta_i^u - \frac{1}{n-1} \sum_{i=1}^{n-1} \theta_i^u \parallel^2 + 4\hat{\eta}^2 \rho_2^2 \sigma_1^2 (n-1)^2 t + 8\hat{\eta}^2 \sum_{i=1}^{n-1} \mathbb{E} \sum_{l=1}^{t} \parallel (\tilde{e}_i^T \tilde{W}^{t_1+l-1} - \frac{1_{n-1}^T}{n-1}) \hat{H}_{(n-1)}^{l-1} \parallel^2$$

$$\leq 2 \sum_{i=1}^{n-1} \mathbb{E} \parallel \theta_i^u - \frac{1}{n-1} \sum_{i=1}^{n-1} \theta_i^u \parallel^2 + 4\hat{\eta}^2 \rho_2^2 \sigma_1^2 (n-1)^2 t + 8\hat{\eta}^2 \rho_2^2 (n-1) \sum_{l=1}^{t} \underbrace{\sum_{i=1}^{n-1} \mathbb{E} \parallel \nabla f_i(\hat{\theta}_i^{l-1}) \parallel^2}_{B_6} \qquad (57)$$

Bound $B_6$:

$$B_6 = \sum_{i=1}^{n-1} \mathbb{E} \parallel \nabla f_i(\hat{\theta}_i^{l-1}) - \nabla f_i(\frac{1}{n-1} \sum_{i=1}^{n-1} \hat{\theta}_i^{l-1}) + \nabla f_i(\frac{1}{n-1} \sum_{i=1}^{n-1} \hat{\theta}_i^{l-1}) - \nabla f(\frac{1}{n-1} \sum_{i=1}^{n-1} \hat{\theta}_i^{l-1}) +$$

$$\nabla f(\frac{1}{n-1} \sum_{i=1}^{n-1} \hat{\theta}_i^{l-1}) - \nabla \tilde{f}(\frac{1}{n-1} \sum_{i=1}^{n-1} \hat{\theta}_i^{l-1}) + \nabla \tilde{f}(\frac{1}{n-1} \sum_{i=1}^{n-1} \hat{\theta}_i^{l-1}) \parallel^2$$

$$\leq 4 \sum_{i=1}^{n-1} \mathbb{E} \parallel \nabla f_i(\hat{\theta}_i^{l-1}) - \nabla f_i(\frac{1}{n-1} \sum_{i=1}^{n-1} \hat{\theta}_i^{l-1}) \parallel^2 + 4 \sum_{i=1}^{n-1} \mathbb{E} \parallel \nabla f_i(\frac{1}{n-1} \sum_{i=1}^{n-1} \hat{\theta}_i^{l-1}) - \nabla f(\frac{1}{n-1} \sum_{i=1}^{n-1} \hat{\theta}_i^{l-1}) \parallel^2 +$$

$$4 \sum_{i=1}^{n-1} \mathbb{E} \parallel \nabla f(\frac{1}{n-1} \sum_{i=1}^{n-1} \hat{\theta}_i^{l-1}) - \nabla \tilde{f}(\frac{1}{n-1} \sum_{i=1}^{n-1} \hat{\theta}_i^{l-1}) \parallel^2 + 4 \sum_{i=1}^{n-1} \mathbb{E} \parallel \nabla \tilde{f}(\frac{1}{n-1} \sum_{i=1}^{n-1} \hat{\theta}_i^{l-1}) \parallel^2$$

$$\leq 4 \sum_{i=1}^{n-1} \mathbb{E} \parallel \nabla f_i(\hat{\theta}_i^{l-1}) - \nabla f_i(\frac{1}{n-1} \sum_{i=1}^{n-1} \hat{\theta}_i^{l-1}) \parallel^2 + 4 \sum_{i=1}^{n-1} \mathbb{E} \parallel \nabla f_i(\frac{1}{n-1} \sum_{i=1}^{n-1} \hat{\theta}_i^{l-1}) - \nabla f(\frac{1}{n-1} \sum_{i=1}^{n-1} \hat{\theta}_i^{l-1}) \parallel^2 +$$

$$4 \sum_{i=1}^{n-1} \mathbb{E} \parallel \frac{1}{n-1} (\nabla f_n(\frac{1}{n-1} \sum_{i=1}^{n-1} \hat{\theta}_i^{l-1}) - \frac{1}{n} \sum_{i=1}^{n} \nabla f_i(\frac{1}{n-1} \sum_{i=1}^{n-1} \hat{\theta}_i^{l-1})) \parallel^2 + 4 \sum_{i=1}^{n-1} \mathbb{E} \parallel \nabla \tilde{f}(\frac{1}{n-1} \sum_{i=1}^{n-1} \hat{\theta}_i^{l-1}) \parallel^2$$

$$\leq 4L^2 \sum_{i=1}^{n-1} \mathbb{E} \parallel \hat{\theta}_i^{l-1} - \frac{1}{n-1} \sum_{i=1}^{n-1} \hat{\theta}_i^{l-1} \parallel^2 + 4(n-1)\sigma_2^2 + \frac{4\sigma_2^2}{n-1} + 4 \sum_{i=1}^{n-1} \mathbb{E} \parallel \nabla \tilde{f}(\frac{1}{n-1} \sum_{i=1}^{n-1} \hat{\theta}_i^{l-1}) \parallel^2 \qquad (58)$$

Substitute Equation (58) into Equation (57) to get

$$\sum_{i=1}^{n-1} \mathbb{E} \parallel \hat{\theta}_i^t - \frac{1}{n-1} \sum_{i=1}^{n-1} \hat{\theta}_i^t \parallel^2$$

$$\leq 2 \sum_{i=1}^{n-1} \mathbb{E} \parallel \theta_i^u - \frac{1}{n-1} \sum_{i=1}^{n-1} \theta_i^u \parallel^2 + 4\hat{\eta}^2 \rho_2^2 \sigma_1^2 (n-1)^2 t + 32\hat{\eta}^2 \rho_2^2 \sigma_2^2 (n-1)^2 t + 32\hat{\eta}^2 \rho_2^2 \sigma_2^2 t +$$

$$32\hat{\eta}^2 L^2 \rho_2^2 (n-1) \sum_{l=1}^{t} \sum_{i=1}^{n-1} \mathbb{E} \parallel \hat{\theta}_i^{l-1} - \frac{1}{n-1} \sum_{i=1}^{n-1} \hat{\theta}_i^{l-1} \parallel^2 + 32\hat{\eta}^2 \rho_2^2 (n-1)^2 \sum_{l=1}^{t} \mathbb{E} \parallel \nabla \tilde{f}(\frac{1}{n-1} \sum_{i=1}^{n-1} \hat{\theta}_i^{l-1}) \parallel^2 \qquad (59)$$

Therefore, it holds that

$$\sum_{t=1}^{T} \sum_{i=1}^{n-1} \mathbb{E} \parallel \hat{\theta}_i^t - \frac{1}{n-1} \sum_{i=1}^{n-1} \hat{\theta}_i^t \parallel^2$$

$$\leq 2T \sum_{i=1}^{n-1} \mathbb{E} \parallel \theta_i^u - \frac{1}{n-1} \sum_{i=1}^{n-1} \theta_i^u \parallel^2 + 4\hat{\eta}^2 \rho_2^2 \sigma_1^2 (n-1)^2 T^2 + 32\hat{\eta}^2 \rho_2^2 \sigma_2^2 (n-1)^2 T^2 + 32\hat{\eta}^2 \rho_2^2 \sigma_2^2 T^2 +$$

$$32\hat{\eta}^2 L^2 \rho_2^2 (n-1)T \sum_{t=0}^{T-1} \sum_{i=1}^{n-1} \mathbb{E} \parallel \hat{\theta}_i^t - \frac{1}{n-1} \sum_{i=1}^{n-1} \hat{\theta}_i^t \parallel^2 + 32\hat{\eta}^2 \rho_2^2 (n-1)^2 T \sum_{t=0}^{T-1} \mathbb{E} \parallel \nabla \tilde{f}(\frac{1}{n-1} \sum_{i=1}^{n-1} \hat{\theta}_i^t) \parallel^2 \quad (60)$$

Then

$$\sum_{t=0}^{T-1} \sum_{i=1}^{n-1} \mathbb{E} \parallel \hat{\theta}_i^t - \frac{1}{n-1} \sum_{i=1}^{n-1} \hat{\theta}_i^t \parallel^2$$

$$\leq 3T \sum_{i=1}^{n-1} \mathbb{E} \parallel \theta_i^u - \frac{1}{n-1} \sum_{i=1}^{n-1} \theta_i^u \parallel^2 + 4\hat{\eta}^2 \rho_2^2 \sigma_1^2 (n-1)^2 T^2 + 32\hat{\eta}^2 \rho_2^2 \sigma_2^2 (n-1)^2 T^2 + 32\hat{\eta}^2 \rho_2^2 \sigma_2^2 T^2 +$$

$$32\hat{\eta}^2 L^2 \rho_2^2 (n-1)T \sum_{t=0}^{T-1} \sum_{i=1}^{n-1} \mathbb{E} \parallel \hat{\theta}_i^t - \frac{1}{n-1} \sum_{i=1}^{n-1} \hat{\theta}_i^t \parallel^2 + 32\hat{\eta}^2 \rho_2^2 (n-1)^2 T \sum_{t=0}^{T-1} \mathbb{E} \parallel \nabla \tilde{f}(\frac{1}{n-1} \sum_{i=1}^{n-1} \hat{\theta}_i^t) \parallel^2 \quad (61)$$

If it satisfies $\hat{\eta} < \sqrt{\frac{1}{32L^2 \rho_2^2 (n-1)T}}$, the following holds:

$$(1 - 32\hat{\eta}^2 L^2 \rho_2^2 (n-1)T) \sum_{t=0}^{T-1} \sum_{i=1}^{n-1} \mathbb{E} \parallel \hat{\theta}_i^t - \frac{1}{n-1} \sum_{i=1}^{n-1} \hat{\theta}_i^t \parallel^2$$

$$\leq 3T \sum_{i=1}^{n-1} \mathbb{E} \parallel \theta_i^u - \frac{1}{n-1} \sum_{i=1}^{n-1} \theta_i^u \parallel^2 + 4\hat{\eta}^2 \rho_2^2 \sigma_1^2 (n-1)^2 T^2 + 32\hat{\eta}^2 \rho_2^2 \sigma_2^2 (n-1)^2 T^2 + 32\hat{\eta}^2 \rho_2^2 \sigma_2^2 T^2 +$$

$$32\hat{\eta}^2 \rho_2^2 (n-1)^2 T \sum_{t=0}^{T-1} \mathbb{E} \parallel \nabla \tilde{f}(\frac{1}{n-1} \sum_{i=1}^{n-1} \hat{\theta}_i^t) \parallel^2 \quad (62)$$

Summing from $t = 1$ to $t = T$ for Equation (56) and substituting Equation (62) into it yields

$$\mathbb{E}\tilde{f}(\frac{1}{n-1} \sum_{i=1}^{n-1} \hat{\theta}_i^T) - \mathbb{E}\tilde{f}(\frac{1}{n-1} \sum_{i=1}^{n-1} \hat{\theta}_i^0)$$

$$\leq \left(\frac{16\hat{\eta}^3 L^2 \rho_2^2 (n-1)T}{1 - 32\hat{\eta}^2 L^2 \rho_2^2 (n-1)T} - \frac{\hat{\eta}}{2}\right) \sum_{t=0}^{T-1} \mathbb{E} \parallel \nabla \tilde{f}(\frac{1}{n-1} \sum_{i=1}^{n-1} \hat{\theta}_i^t) \parallel^2 - \frac{\hat{\eta}(1 - \hat{\eta}L)}{2} \sum_{t=0}^{T-1} \mathbb{E} \parallel \frac{1}{n-1} \sum_{i=1}^{n-1} \nabla f_i(\hat{\theta}_i^{t-1}) \parallel^2 +$$

$$\frac{1}{2(n-1)} \cdot \frac{3\hat{\eta}L^2 T}{1 - 32\hat{\eta}^2 L^2 \rho_2^2 (n-1)T} \sum_{i=1}^{n-1} \mathbb{E} \parallel \theta_i^u - \frac{1}{n-1} \sum_{i=1}^{n-1} \theta_i^u \parallel^2 + \frac{2\hat{\eta}^3 L^2 \rho_2^2 \sigma_1^2 (n-1)T^2}{1 - 32\hat{\eta}^2 L^2 \rho_2^2 (n-1)T} +$$

$$\frac{16\hat{\eta}^3 L^2 \rho_2^2 \sigma_2^2 (n-1)T^2}{1 - 32\hat{\eta}^2 L^2 \rho_2^2 (n-1)T} + \frac{16\hat{\eta}^3 L^2 \rho_2^2 \sigma_2^2 T^2}{(n-1)(1 - 32\hat{\eta}^2 L^2 \rho_2^2 (n-1)T)} + \frac{\hat{\eta}^2 L \sigma_1^2 T}{2(n-1)} \quad (63)$$

where $\hat{\eta} < \sqrt{\frac{1}{32L^2 \rho_2^2 (n-1)T}}$.

Then let

$$D_4 = \frac{1}{2} - \frac{16\hat{\eta}^2 L^2 \rho_2^2 (n-1)T}{1 - 32\hat{\eta}^2 L^2 \rho_2^2 (n-1)T}, \quad D_5 = \frac{1}{2} - \frac{\hat{\eta}L}{2}$$

$$D_6 = 1 - 32\hat{\eta}^2 L^2 \rho_2^2 (n-1)T.$$

If the step size $\hat{\eta}$ satisfies $\hat{\eta} < \sqrt{\frac{1}{32L^2 \rho_2^2 (n-1)T}}$, we have the following convergence result of the subsequent $T$ rounds of training:

$$D_4 \cdot \frac{1}{T} \sum_{t=0}^{T-1} \mathbb{E} \parallel \nabla \tilde{f}(\frac{1}{n-1} \sum_{i=1}^{n-1} \hat{\theta}_i^t) \parallel^2 + D_5 \frac{1}{T} \sum_{t=0}^{T-1} \mathbb{E} \parallel \frac{1}{n-1} \sum_{i=1}^{n-1} \nabla \tilde{f}(\hat{\theta}_i^t) \parallel^2$$

$$\leq \frac{3L^2}{2D_6} \cdot \frac{1}{n-1} \sum_{i=1}^{n-1} \mathbb{E} \parallel \theta_i^u - \frac{1}{n-1} \sum_{i=1}^{n-1} \theta_i^u \parallel^2 + \frac{\hat{\eta} L \sigma_1^2}{2(n-1)} + \frac{\tilde{f}(\frac{1}{n-1} \sum_{i=1}^{n-1} \theta_i^u) - \tilde{f}^*}{\hat{\eta} T} + \frac{2\hat{\eta}^2 L^2 \rho_2^2 \sigma_1^2 (n-1) T}{D_6} +$$

$$\frac{16\hat{\eta}^2 L^2 \rho_2^2 \sigma_2^2 (n-1) T}{D_6} + \frac{16\hat{\eta}^2 L^2 \rho_2^2 \sigma_2^2 T}{(n-1) D_6}$$

which completes the proof.

## H. Proof of Corollary 4.12

We assume that $D_4 \geq C$, where $C \in (0, \frac{1}{2})$. Then it holds

$$\frac{16\hat{\eta}^2 L^2 \rho_2^2 (n-1) T}{1 - 32\hat{\eta}^2 L^2 \rho_2^2 (n-1) T} = \frac{1}{2} - D_4 \leq \frac{1 - 2C}{2} \Leftrightarrow \hat{\eta}^2 \leq \frac{1 - 2C}{64(1-C) L^2 \rho_2^2 (n-1) T}$$

$$\text{and} \quad D_6 = 1 - 32\hat{\eta}^2 L^2 \rho_2^2 (n-1) T \geq \frac{1}{2(1-C)}$$

If we set $\hat{\eta} = \frac{n-1}{T}$, the following should be satisfied

$$\frac{(n-1)^2}{T^2} \leq \frac{1 - 2C}{64(1-C) L^2 \rho_2^2 (n-1) T} \Leftrightarrow T \geq \frac{64(1-C) L^2 \rho_2^2 (n-1)^3}{1 - 2C}$$

$$D_5 = \frac{1}{2} - \frac{\hat{\eta} L}{2} = \frac{1}{2} - \frac{(n-1) L}{2T} \geq 0 \Leftrightarrow T \geq (n-1) L$$

As a result, it holds that

$$\begin{cases} \dfrac{\tilde{f}(\frac{1}{n-1} \sum_{i=1}^{n-1} \theta_i^u) - \tilde{f}^*}{\hat{\eta} T} = \dfrac{\tilde{f}(\frac{1}{n-1} \sum_{i=1}^{n-1} \theta_i^u) - \tilde{f}^*}{n-1} \\[4mm] \dfrac{\hat{\eta} L \sigma_1^2}{2(n-1)} = \dfrac{L \sigma_1^2}{2T} \\[4mm] \dfrac{2\hat{\eta}^2 L^2 \rho_2^2 \sigma_1^2 (n-1) T}{D_6} = (\frac{1}{2} - D_4) \cdot \dfrac{\sigma_1^2}{8} \leq \dfrac{(1-2C)\sigma_1^2}{16} \\[4mm] \dfrac{16\hat{\eta}^2 L^2 \rho_2^2 \sigma_2^2 (n-1) T}{D_6} = (\frac{1}{2} - D_4) \cdot \sigma_2^2 \leq \dfrac{(1-2C)\sigma_2^2}{2} \\[4mm] \dfrac{16\hat{\eta}^2 L^2 \rho_2^2 \sigma_2^2 T}{(n-1) D_6} = (\frac{1}{2} - D_4) \cdot \dfrac{\sigma_2^2}{(n-1)^2} \leq \dfrac{(1-2C)\sigma_2^2}{2(n-1)^2} \end{cases}$$

Therefore, if the number of training round $T$ satisfies $T \geq \max\{\frac{64(1-C) L^2 \rho_2^2 (n-1)^3}{1 - 2C}, (n-1) L\}$ with $C \in (0, \frac{1}{2})$, and the step size $\hat{\eta}$ further satisfies $\hat{\eta} = \frac{n-1}{T}$, then Corollary 4.12 holds.

## I. Experimental Details

### I.1. Datasets and Models

We train the ResNet-18 model (He et al., 2016) and the CNN model on the real-world datasets, including MNIST (Lecun et al., 1998), CIFAR-10 (Krizhevsky & Hinton, 2009), SVHN (Netzer et al., 2011) and Fashion-MNIST (Xiao et al., 2017).

- **MNIST** is a handwritten digit dataset containing $60,000$ training images and $10,000$ test images of grayscale digits $(0-9)$, each with a resolution of $28 \times 28$ pixels.

- **Fashion-MNIST** consists of $60,000$ training images and $10,000$ testing images, with each being a $28 \times 28$ pixel grayscale image representing various clothing items such as T-shirts, pants, dresses, and more.

- **CIFAR-10** includes $60,000$ colored images of 10 common object classes, with $50,000$ images for training and $10,000$ for testing, each at $32 \times 32$ pixels with three color channels (RGB).

- **SVHN** consists of real-world house number images from street views, containing $73,257$ training images and $26,032$ test images, also at $32 \times 32$ pixels with RGB channels.

### I.2. Baseline methods

To show the advantages of our proposed algorithm, we compare it with the following baseline methods:

- **Origin.** The baseline is actually the D-PSGD algorithm (Lian et al., 2017), which does not involve any unlearning operations.

- **Retrain.** This method retrains the decentralized models from scratch on the remaining $n-1$ clients after removing the target client-$n$. It can achieve exact unlearning but requires significant time and resources.

- **FATS-Unl (Tao et al., 2024).** It saves the historical global models and the sets of clients that participated in past training rounds on the central server. When a client initiates an unlearning request, the algorithm retrieves the latest model from before the client's initial participation in training and then retrains.

- **FedRecovery (Zhang et al., 2023).** It relies on a central server to remove the weighted sum of the gradient residuals from the global model to eliminate the influence of a certain client, and adds specific Gaussian noise to make the unlearned model and the retrained model statistically indistinguishable.

- **HDUS (Ye et al., 2024).** Each client's main model relies entirely on local and public datasets and remains unaffected by its neighbors. The collaboration among clients is solely reflected in the integration of distilled seed models from neighbors to make decisions. Therefore, the decision results of the client initiating the unlearning request can be removed by adjusting the integration process.

### I.3. Metrics

In our experiments, we use multiple metrics to evaluate the performance of the proposed decentralized unlearning algorithm, including accuracy, unlearning time, communication overhead, and attack success rate.

- **Accuracy.** To examine the statistical indistinguishability, we evaluate the overall accuracy of the unlearning models and the retrained models. To show the effectiveness of our PDUDT, we record the average accuracy on each class after performing a specified number of rounds.

- **Unlearning time.** We evaluate the running time required for the proposed Algorithm 2 to perform the unlearning operations and compare it with other baseline algorithms. For FATS-Unl and FedRecovery, this is measured on the server side, while for others, it is tracked on the client with the most neighbors.

- **Attack success rate.** Membership Inference Attack (MIA) is employed to determine whether the data samples of a client slated for forgetting were part of the training process. A higher MIA success rate indicates that the global model still retains considerable information about this client's training data, signifying an inadequate unlearning effect. In contrast, a success rate of 50%—equivalent to random guessing—implies that the model no longer carries exploitable traces of the client's data, thereby demonstrating effective client removal. We perform the membership inference attack on the unlearned model to verify if the proposed unlearning algorithm successfully removed the targeted client's influence.

### I.4. Additional experimental results

To show the scalability of PDUDT, we conducted experiments with 20 clients performing a natural language processing (NLP) task. Specifically, we evaluated PDUDT using the Yahoo! Answers dataset with the Bert-tiny model.

For the Yahoo! Answers dataset, Figure 3 demonstrates that our method is statistically indistinguishable from the Perturbed Retraining approach across a range of noise scales. Moreover, Figure 4 highlights the effectiveness of both PDUDT and

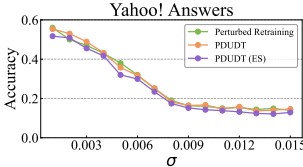

*Figure 3.* The accuracy of unlearned models using PDUDT, PDUDT (ES), and perturbed retrained models on Yahoo! Answers dataset.

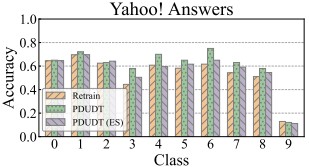

*Figure 4.* The accuracy on each class using PDUDT, PDUDT (ES), and perturbed retrained models on Yahoo! Answers dataset.

*Table 5.* Comparison of the unlearning time and the attack precision of MIA across different unlearning methods on Yahoo! Answers dataset. "-" means no results or not applicable.

| Method | Unlearning time (s) | Attack precision (%) |
|---|---|---|
| Origin | - | $69.2 \pm 0.2$ |
| Retrain | 2852.2 | $50.3 \pm 0.8$ |
| FATS-Unl | 2039.9 | $51.2 \pm 0.4$ |
| FedRecovery | 13.0 | $51.8 \pm 0.7$ |
| HDUS | 27.5 | $51.5 \pm 0.3$ |
| **PDUDT** | 11.8 | $50.7 \pm 0.7$ |
| **PDUDT (ES)** | 10.2 | $51.1 \pm 0.5$ |

its space-efficient variant, PDUDT (ES): While they maintain high performance on classes 0–8, their accuracy noticeably declines for Class 9. Finally, Table 5 summarizes the substantial time savings afforded by our unlearning operations and further confirms the unlearning effectiveness of PDUDT and PDUDT (ES) through comparable membership inference attack (MIA) precision relative to the Retrain method. The results confirm the superior performance of PDUDT, showing good scalability to larger networks and NLP task.

