# OpenReview forum: "PDUDT: Provable Decentralized Unlearning under Dynamic Topologies"
_ICML.cc/2025/Conference — ICML 2025 poster_

### Official Review · Reviewer_6f9d · 2025-03-10

**Overall Recommendation:** 4

**Summary:**

The paper proposes PDUDT, a Provable Decentralized Unlearning algorithm under Dynamic Topologies. The key contribution is a decentralized method that eliminates the influence of a specific client without additional communication or retraining. The authors provide rigorous theoretical guarantees demonstrating that PDUDT is statistically indistinguishable from perturbed retraining and achieves a convergence rate of O(1/T), matching state-of-the-art results. Experimental results show that PDUDT saves over 99% unlearning time compared to retraining while maintaining comparable unlearning performance.

**Claims And Evidence:**

The claims made in the paper are largely supported by theoretical analysis, experimental results, and comparative evaluations. Here’s an assessment of the key claims and their supporting evidence:
1. PDUDT enables decentralized unlearning without additional communication or retraining: The proposed algorithm uses historical gradient information to approximate gradient residuals and eliminate a client’s contribution without retraining. The method does not require extra communication beyond standard decentralized learning updates. The algorithm’s efficiency is demonstrated theoretically and empirically, showing that it performs unlearning in O(t1) time, significantly reducing computational overhead compared to retraining-based methods.
2. PDUDT is statistically indistinguishable from the perturbed retrained method: The paper provides a rigorous proof that PDUDT satisfies (epsilon, beta)-machine unlearning, ensuring that the output model is statistically close to one that would have been obtained by retraining from scratch. A Gaussian noise mechanism is added to further enhance indistinguishability, following the principles of differential privacy.
3. PDUDT effectively removes the influence of the unlearned client: The membership inference attack (MIA) evaluation shows that after unlearning, the attack success rate drops to approximately 50%, indicating that the model has effectively "forgotten" the removed client’s data. The accuracy on the unlearned client’s class drops significantly, while accuracy on other classes remains stable, reinforcing that the client's influence has been successfully removed.

**Essential References Not Discussed:**

The paper provides a comprehensive discussion of prior work in federated/decentralized unlearning, there do not appear to be critical missing references that would significantly alter the context or evaluation of the paper’s contributions. The theoretical and experimental analysis is well-grounded in previous research, making the discussion of related work sufficient and appropriate for the scope of the study.

**Experimental Designs Or Analyses:**

The experimental designs and analyses in this paper appear well-structured and methodologically sound. The paper employs a variety of evaluation metrics and benchmark datasets to assess the effectiveness of the proposed PDUDT algorithm. Below is an assessment of the soundness and validity of these experiments:
1. The experiments are conducted on standard benchmark datasets: MNIST, Fashion-MNIST, CIFAR-10, and SVHN, ensuring generalizability. The models used (CNN for simpler datasets and ResNet-18 for more complex datasets) align well with the dataset complexity. The experiments simulate decentralized environments with dynamic topologies, making the setup realistic for real-world applications. PDUDT is compared against multiple baseline methods (Retrain, FATS-Unl, FedRecovery, and HDUS), allowing for a comprehensive performance evaluation.
While the decentralized learning setup is reasonable, the details on how the topology evolves dynamically over time are limited. More information on how client connections change could improve reproducibility.

2. The paper evaluates PDUDT using three primary metrics:
— Accuracy: To assess whether the model successfully forgets the unlearned data while maintaining overall performance.
— Unlearning Time: To measure computational efficiency, showing that PDUDT is significantly faster than retraining-based methods.
— Attack Resistance: Using membership inference attacks (MIA) to verify if the target client’s data has truly been forgotten.
These metrics effectively capture both efficiency and security aspects of unlearning. However, the impact of different levels of noise injection (used for statistical indistinguishability) on model utility and unlearning effectiveness could be further analyzed.

In summary, the experimental design is methodologically sound, and the analyses provide strong empirical evidence supporting PDUDT’s effectiveness. While the paper presents robust results, additional details on topology evolution and noise impact could further enhance validity.

**Methods And Evaluation Criteria:**

This paper effectively tackles the challenge of data unlearning in decentralized machine learning environments. The proposed methods are well-structured, and the evaluation criteria are comprehensive. Here’s why:
1. The paper focuses on decentralized unlearning under dynamic topologies, a complex and underexplored issue. The proposed PDUDT introduces a gradient residual approximation technique that enables unlearning without retraining or additional communication, ensuring scalability and efficiency. Theoretical analysis guarantees that PDUDT is statistically indistinguishable from retraining-based unlearning.
2. The paper employs multiple experimental metrics to assess the performance of the proposed unlearning algorithm, including:
— Accuracy: Ensuring that the model retains high performance on unaffected data while effectively forgetting the targeted data.
— Unlearning Time: Demonstrating that PDUDT achieves significant efficiency gains compared to retraining-based methods.
— Attack Resistance: Validating unlearning effectiveness using membership inference attacks (MIA) to confirm that the removed data is no longer recoverable.

Overall, the paper provides a well-founded, efficient, and theoretically sound approach to decentralized unlearning, supported by rigorous evaluation and empirical evidence.

**Other Comments Or Suggestions:**

1. Given the extensive use of mathematical symbols throughout the paper, a notation table summarizing key variables and parameters could improve readability and accessibility.
2. Since the appendix contains many sections with detailed proofs and experimental setups, it would be helpful to add a brief overview at the beginning of the appendix summarizing the role of each section. This would make it easier for readers to navigate the supplementary material.
3. A careful proofreading of the paper may help identify and correct any minor spelling or grammatical errors to ensure clarity and professionalism.

**Other Strengths And Weaknesses:**

Strengths:
1. PDUDT is the first provable decentralized unlearning algorithm that operates under dynamic topologies, addressing a significant gap in privacy-preserving decentralized learning.
2. The theoretical results of this article are detailed and in-depth. The paper derives an upper bound on the deviation between PDUDT’s output and a retrained model, providing rigorous guarantees on its unlearning performance (Theorem 4.9). The convergence proof (Theorem 4.11) ensures that after unlearning, the decentralized learning process continues to perform optimally, matching state-of-the-art decentralized learning results.
3. The experimental section is comprehensive, covering multiple datasets and comparing against strong baselines. The performance advantages of the proposed PDUDT are demonstrated from multiple perspectives.
4. The paper is well-structured, with clear theoretical formulations, algorithm descriptions, and proofs that make the methodology easy to follow.


Weaknesses:
1. Some experimental details are missing. For example, the paper does not explicitly clarify whether the experiments fully simulate dynamic network connections, which is crucial for demonstrating the adaptability of PDUDT in real-world decentralized settings.
2. While the theoretical analysis is rigorous, some notations in the proofs are complex and could be slightly simplified for readability, especially for readers less familiar with decentralized optimization.

**Questions For Authors:**

1. How is network dynamism reflected in the theoretical results? The paper discusses dynamic topologies, but could you clarify where this is explicitly incorporated into the theoretical analysis?
2. Additionally, was the dynamic nature of network links simulated in the experiments? If so, how frequently were connections updated or changed?

**Relation To Broader Scientific Literature:**

The key contributions of PDUDT relate to several areas of research, including machine unlearning, decentralized learning, efficient model updates, and theoretical guarantees. Prior works in machine unlearning, such as FedEraser (Liu et al., 2021) and FedRecovery (Zhang et al., 2023), primarily focus on removing client contributions in federated learning, often relying on gradient residual removal or retraining, but they assume the presence of a central server, limiting their applicability to fully decentralized systems. Recent decentralized unlearning approaches, such as HDUS (Ye et al., 2024) and BlockFUL (Liu et al., 2024a), propose heuristic solutions using distilled models or blockchain structures, but they lack rigorous theoretical guarantees. PDUDT addresses these limitations by introducing the first provable decentralized unlearning method, leveraging gradient residual approximations and a weighted removal mechanism to eliminate a client’s influence without requiring retraining or additional communication overhead, making it highly efficient in dynamic decentralized networks.

**Theoretical Claims:**

The theoretical claims in the paper are well-supported and logically consistent, following standard mathematical frameworks from differential privacy and decentralized optimization. The proofs provide strong guarantees on statistical indistinguishability and convergence:
1. The (epsilon, beta)-machine unlearning guarantee is based on differential privacy principles, ensuring that the model after unlearning is statistically indistinguishable from a retrained model. The proof constructs an upper bound on the difference between PDUDT’s output and a fully retrained model and then introduces Gaussian noise to mask this difference. By leveraging the Gaussian mechanism theorem, the authors establish that no one can reliably distinguish between the two models within a given probability bound.
2. The convergence analysis aims to prove that PDUDT maintains a O(1/T) convergence rate after an unlearning operation. The proof follows a decentralized gradient descent approach, deriving an upper bound on the expected gradient norm over T communication rounds. By incorporating assumptions of Lipschitz smoothness and bounded variance, the analysis confirms that the learning process continues to converge at the same rate as standard decentralized optimization methods. The step-size conditions are carefully derived to ensure stability.

---

> ### Author Rebuttal · Authors · 2025-03-31
>
> We thank the Reviewer 6f9d for the time and valuable feedback! We would try our best to address the comments one by one.
>
> **1. Response to “Experimental Designs Or Analyses”:**
> Thank you for your constructive feedback. We agree that providing more details on topology evolution and the impact of noise would further strengthen our results. In the experiments, we randomly generate connections among clients and then assign communication weights using the Metropolis-Hastings method. More details are provided in our subsequent response (Response to “Other Weaknesses 1.” ＆ “Questions For Authors 2.”). As for the impact of noise, our Figure 1 demonstrates that increased noise leads to a decline in model accuracy. However, under all noise conditions, the performance of PDUDT and Perturbed Retraining method is always comparable. This indicates that under the same noise conditions, these two approaches can achieve statistical indistinguishability. We hope our response addresses your concerns.
>
>
> **2. Response to “Other Weaknesses 1.” ＆ “Questions For Authors 2.”:**
> Thank you for your constructive feedback. In fact, our experiments are based on dynamic connections. In each round, whether there is a connection between any two clients is randomly generated. Then, in order to ensure that the communication situation can be modeled as a doubly stochastic matrix, we use the Metropolis-Hastings method cited in the paper to generate the communication weights between clients. We have provided this detail of dynamic topology construction in the revised version.
>
>
> **3. Response to “Other Weaknesses 2.” ＆ “Other Comments Or Suggestions 1.”:**
> Thank you for your valuable feedback. We agree that the extensive use of mathematical symbols can be challenging for readers. In our revised version, we have added a comprehensive notation table that summarizes all key variables and parameters, which enhance both readability and accessibility. Thank you again!
>
> **4. Response to “Questions For Authors 1.”:**
> Thank you for your insightful question. Network dynamism is explicitly modeled in our theoretical analysis by allowing the communication matrices $\mathcal{W}^t$ (and the corresponding retraining matrices $\tilde{\mathcal{W}}^t$ ) to be time-varying. This reflects the fact that the network topology can change at each round. We introduce spectral gap upper bounds (e.g., $\rho_1$ and $\rho_2$) in our Assumption 4.8, which quantify the connectivity properties of these matrices. These bounds directly influence the convergence rate and error bounds derived in Theorem 4.9 and subsequent results. We hope our answer can clarify your confusion.
>
>
> **5. Response to “Other Comments Or Suggestions 2.＆3.”:**
> We thank the reviewer for these valuable suggestions. In response, we have added a concise overview at the beginning of the appendix that outlines the contents and purpose of each section, making it easier for readers to navigate the supplementary material. Additionally, we carefully proofread the entire paper to correct any minor spelling or grammatical errors, ensuring clarity and a professional presentation. We hope these improvements meet your expectations.
>
> If there are any further confusions/questions, we are happy to clarify and try to address them. Thank you again and your recognition means a lot for our work.

---

### Official Review · Reviewer_vxha · 2025-03-13

**Overall Recommendation:** 3

**Summary:**

The authors introduce a novel algorithm PDUDT, which is designed to enable efficient and provable unlearning in decentralized learning systems with dynamic network topologies. PDUDT allows clients to remove the influence of a specific client without retraining or additional communication by using historical gradient submissions to approximate gradient residuals and adjusting local models accordingly. Theoretical guarantees show PDUDT achieves $\mathcal{O}(\frac{1}{T})$  convergence rate. Experiments on datasets like MNIST, CIFAR-10, and SVHN demonstrate that PDUDT reduces unlearning time by over 99% compared to retraining.

**Claims And Evidence:**

This work is generally supported by rigorous theoretical analysis. However, I still have some concerns:
(1) The authors claim that they are the first provable decentralized unlearning algorithm under dynamic topologies. Some related works, such as “Decentralized Federated Unlearning on Blockchain, arxiv 2024; Decentralized Unlearning for Trustworthy AI-Generated Content (AIGC) Services, IEEE Network 2024; Heterogeneous Decentralized Machine Unlearning with Seed Model Distillation, arxiv 2023” need to be carefully addressed and compared.
(2) The claim that PDUDT is statistically indistinguishable from perturbed retraining, the authors utilize Gaussian mechanism to achieve it. However, the connection between the Gaussian noise and the indistinguishability guarantee could be explained more clearly.

**Essential References Not Discussed:**

Some decentralized unlearning algorithms have also been proposed, such as “Decentralized Federated Unlearning on Blockchain, arxiv 2024; Decentralized Unlearning for Trustworthy AI-Generated Content (AIGC) Services, IEEE Network 2024; Heterogeneous Decentralized Machine Unlearning with Seed Model Distillation, arxiv 2023”, what are the main contributions of PDUDT?

**Experimental Designs Or Analyses:**

(1) While the paper demonstrates the effectiveness of PDUDT on a network of 10 clients, the scalability to much larger networks (e.g., hundreds of clients) is not thoroughly explored.
(2) The experiments are conducted on specific models and datasets. While the results are convincing for these cases, the generalizability to other models and tasks (e.g., natural language processing or more complex architectures) is not fully addressed.

**Methods And Evaluation Criteria:**

The chosen baselines and the evaluation criteria are well-suited for the decentralized unlearning problems.

**Other Comments Or Suggestions:**

None.

**Other Strengths And Weaknesses:**

While the early stopping strategy is introduced to save memory, the paper does not provide a detailed analysis of its impact on unlearning performance. A more thorough discussion of the trade-offs between memory savings and performance would enhance clarity.

**Questions For Authors:**

(1) Some decentralized unlearning algorithms, such as “Decentralized Federated Unlearning on Blockchain, arxiv 2024; Decentralized Unlearning for Trustworthy AI-Generated Content (AIGC) Services, IEEE Network 2024; Heterogeneous Decentralized Machine Unlearning with Seed Model Distillation, arxiv 2023”, need to be addressed.
(2) While the paper demonstrates the effectiveness of PDUDT on a network of 10 clients, the scalability to much larger networks (e.g., hundreds of clients) is not thoroughly explored. The experiments are conducted on specific models and datasets. While the results are convincing for these cases, the generalizability to other models and tasks (e.g., natural language processing or more complex architectures) is not fully addressed.

**Relation To Broader Scientific Literature:**

Theoretical guarantees for unlearning have been explored in centralized settings, with methods like Exact Unlearning (Guo et al., 2020) and Approximate Unlearning (Sekhari et al., 2021) providing formal guarantees for removing data points from trained models. PDUDT provides provable guarantees for decentralized unlearning, including statistical indistinguishability and convergence rates. This extends prior theoretical work to the decentralized setting, where the lack of a central server introduces additional challenges.

**Theoretical Claims:**

Almost check the correctness, I have questions about the Corollary 4.10, The choice of the noise scale \sigma is based on the upper bound from Theorem 4.9. If the bound in Theorem 4.9 is not accurate, the noise scale might not be appropriate, and the statistical indistinguishability claim could be affected. Also, in practice, the Gaussian mechanism's effectiveness might be limited by the actual distribution of the data and model parameters.

---

> ### Author Rebuttal · Authors · 2025-04-01
>
> We thank the Reviewer vxha for the valuable feedback! We would try our best to address the comments one by one.
>
> **Response to “Claims And Evidence (1)” ＆ “Essential References Not Discussed” ＆ “Questions For Authors (1)”:**
> We have carefully examined these works and provide a detailed comparison below to clarify our unique contributions.
> **(1) vs. HDUS (Ye et al., 2023):** HDUS relies on seed model distillation and additional training steps, without theoretical guarantees. PDUDT, by contrast, eliminates influence using saved historical gradients—without retraining—and offers formal guarantees on unlearning and convergence under dynamic topology.
> **(2) vs. AIGC Unlearning (Lin et al., 2024):** The AIGC work focuses on privacy-preserving AIGC systems via coded computing, incurring storage and reconstruction overhead. It lacks theoretical unlearning guarantees and is tailored to AIGC tasks. In contrast, PDUDT is application-agnostic, retraining-free, and provides provable client-level unlearning.
> **(3) vs. BlockFUL (Liu et al., 2024):** BlockFUL depends on blockchain infrastructure and consensus, which may be impractical in dynamic or asynchronous networks. In contrast, PDUDT is lightweight and naturally supports dynamic peer-to-peer topologies.
> Therefore, we claim that our work presents the first **provable** decentralized unlearning algorithm under **dynamic topologies**.
> In our current manuscript, although HDUS and BlockFUL have been cited, we agree that adding more detailed discussion will improve the quality. More generally, we also fill in the missing discussion on AIGC Unlearning (Lin et al., 2024).
>
> **Response to “Claims And Evidence (2)”:**
> The statistical indistinguishability guarantee in Corollary 4.10 relies on the fact that the difference between the PDUDT output (model corrected by gradient residual estimation) and the output of retraining can be bounded (as shown in Theorem 4.9), and then obfuscated by adding Gaussian noise with a corresponding scale. Specifically, the Gaussian mechanism ensures that, after adding noise drawn from $\mathcal{N}(0,\sigma^2\mathbb{I}_d)$, the two outputs become close in distribution—making them statistically indistinguishable.
> This approach follows the standard principles of the Gaussian mechanism in differential privacy. In our case, Theorem 4.9 gives an upper bound on this difference, and we use it to determine a noise scale, thereby masking any deviation between the PDUDT output and the retraining output.
>
> **Response to “Theoretical Claims”:** We fully understand your concern about the accuracy of the difference between the retrained and the unlearned models, as it affects the subsequent noise scale. But in fact, we cannot get the exact value because $\tilde{\theta}_i^{t_1}$ is a retrained model. Therefore, we need to approximately seek the upper bound of it through the model iteration rule. Our Theorem 4.9 provides a conservative upper bound on the difference between the retrained and the unlearned models, which ensures that the noise scale $\sigma$ selected suffices to achieve the desired level of statistical indistinguishability.
> While the bound might not be tight in some scenarios, this conservativeness is intentional and aligns with the standard approach in differential privacy, where upper bounds are used to ensure worst-case guarantees. We also highlight that the Gaussian mechanism is chosen due to its robustness under such uncertainty—it provides meaningful guarantees as long as the upper bound is valid.
>
> **Response to “Experimental Designs Or Analyses” ＆ “Questions For Authors (2)”:**
> Due to limited computational resources and funding constraints, we are currently unable to simulate scenarios involving hundreds of clients. However, to show the scalability, we conducted experiments with 20 clients on a NLP task. Specifically, we evaluated PDUDT using the Yahoo! Answers dataset with the Bert-tiny model. The results can be found at: https://anonymous.4open.science/r/Unlearning-47E3, confirming that PDUDT has good scalability to larger networks and NLP task.
>
> **Response to “Other Strengths And Weaknesses”:** We would like to clarify that Corollary 4.10 in our paper also applies to the early stopping variant PDUDT (ES). The key difference between them lies in their cumulative weighted gradient residual approximation ($\frac{1}{n-1}\sum\limits_{i=1}^{n-1}\sum\limits_{t=0}^{t_1-1}p_i^t \delta_i^t$ for PDUDT and $\frac{1}{n-1}\sum\limits_{i=1}^{n-1}\sum\limits_{t=0}^{t_{1,i}-1}p_i^{'t} \delta_i^t $ for PDUDT (ES)). Despite it, they both can be bounded by the same upper bound in Theorem 4.9. This is formally justified in Appendix A (Lemma A.2) and further supported by the derivations in Appendix C. We note that the current manuscript did not explicitly state that Corollary 4.10 also holds for PDUDT (ES). We have clarified this in the revised version.
>
> If you have further confusions, we are happy to clarify them. Thank you again for your support.

---

### Official Review · Reviewer_rNoR · 2025-03-13

**Overall Recommendation:** 3

**Summary:**

This work focuses on the provable unlearning in decentralized learning under dynamic topologies. The proposed PDUDT algorithm addresses this by using historical gradient information of clients and their neighbors to eliminate a specific client's influence without extra communication or retraining. The authors provide rigorous theoretical guarantees, showing its statistical indistinguishability from perturbed retraining and $\(O(\frac{1}{T})\)$ convergence rate.

**Claims And Evidence:**

The claim that PDUDT can eliminate the influence of a specific client without additional communication or retraining is supported by the algorithm design. The algorithm uses historical gradient submissions to compute gradient residual approximations and weights, which are then used to adjust the local models. However, the approximation of gradient residuals may not fully capture the complex interactions among clients in all cases. This could potentially lead to incomplete elimination of the client's influence, especially in scenarios with highly non-linear model dynamics.

**Essential References Not Discussed:**

I think the highly related works have been cited and discussed.

**Experimental Designs Or Analyses:**

Yes, The paper's experimental designs and analyses generally show soundness. The selection of datasets (MNIST, FashionMNIST, CIFAR - 10, and SVHN) ,  models (CNN and ResNet - 18), the baselines are appropriate. But I do not understand the Attack precision design, which need more details.

**Methods And Evaluation Criteria:**

Yes, I think the experiments are sufficient, the proposed methods and evaluation criteria in the paper are highly relevant and make sense.

**Other Comments Or Suggestions:**

No any other comments or suggestions, overall, I think this work is well-written.

**Other Strengths And Weaknesses:**

No any other comments or suggestions.

**Questions For Authors:**

My major concern is still about the PDUDT algorithm design, the claim that PDUDT can eliminate the influence of a specific client without additional communication or retraining is supported by the algorithm design. The algorithm uses historical gradient submissions to compute gradient residual approximations and weights, which are then used to adjust the local models. However, the approximation of gradient residuals may not fully capture the complex interactions among clients in all cases. This could potentially lead to incomplete elimination of the client's influence, especially in scenarios with highly non-linear model dynamics. If this can be well addressed, I can consider changing my evaluation.

**Relation To Broader Scientific Literature:**

The key contribution is that this work is the first provable decentralized unlearning algorithm under dynamic topologies. The concept of (\epsilon, \beta)-indistinguishability (Neel et al., 2021) used in the paper to prove that PDUDT is statistically indistinguishable from perturbed retraining. This builds on the existing literature on differential privacy in distributed learning.

**Theoretical Claims:**

The proof assumes that the communication matrices are doubly stochastic and symmetric (Assumption 4.7). While this is a common assumption, it may not hold in all real-world decentralized systems, especially under dynamic topologies. A discussion of how the proof generalizes to non-symmetric or non-doubly stochastic matrices would be beneficial.

---

> ### Author Rebuttal · Authors · 2025-03-31
>
> We thank the Reviewer rNoR for the time and valuable feedback! We would try our best to address the comments one by one.
>
> **1. Response to “Claims And Evidence”＆ “Questions For Authors”:**
> Thank you for your insightful feedback. As we discussed in Related Work, approximate unlearning  has demonstrated sufficient effectiveness in many practical scenarios and has thus motivated further work by many researchers, such as Liu et al. (2021) and Ye et al. (2024). Therefore, in this work, we primarily employ the weighted gradient residual approximation to adjust the local model, further achieving approximate unlearning in the decentralized paradigm. Corollary 4.10 shows the unlearning performance that our algorithm can achieve, which is statistically indistinguishable from perturbed retraining. In addition to the theoretical guarantee, we experimentally verify its statistical indistinguishability and further analyze the utility and efficiency of PDUDT. We hope our answer can clarify your confusion.
>
>
>
> **2. Response to “Theoretical Claims”:**
> Thank you for your valuable feedback. We note that Assumption 4.7 is a standard and widely adopted assumption in decentralized learning frameworks. Furthermore, many methods have been proposed to construct a doubly stochastic communication matrix, such as the Metropolis-Hastings method. However, under extremely harsh communication conditions, the use of Assumption 4.7 has certain limitations, since it may be challenging to guarantee the doubly stochastic nature of the communication matrix. We have included a discussion on this limitation in the revised version. Thank you once again for your insightful comments.
>
>
>
> **3. Response to “Experimental Designs Or Analyses”:**
> Thank you for your insightful feedback. We hope that the following explanation of MIA design can clarify your confusion: Membership Inference Attack (MIA) is employed to determine whether the data samples of a client slated for forgetting were part of the training process. A higher MIA success rate indicates that the global model still retains considerable information about this client's training data, signifying an inadequate unlearning effect. In contrast, a success rate of 50%—equivalent to random guessing—implies that the model no longer carries exploitable traces of the client's data, thereby demonstrating effective client removal. Thank you again for your valuable comments, we have added this explanation in the revised version.
>
> If there are any further confusions/questions, we are happy to clarify and try to address them. Thank you again and your recognition means a lot for our work.

---

### Decision · Program_Chairs · 2025-05-01

**Decision:**

Accept (poster)

**Comment:**

This paper proposes a provable decentralised unlearning approach. The reviewers found this paper (1) proposes an interesting and meaningful algorithm, (2) the experiments are comprehensive and solid, (3) the algorithm is supported by solid theory, etc. I agree with the authors. So, I recommend to accept this paper.